



# Anthropogenic activities significantly increase annual greenhouse gas (GHG) fluxes from temperate headwater streams in Germany

**Authors:** Ricky Mwangada Mwanake[1]; Gretchen Maria Gettel[2], Elizabeth Gachibu Wangari[1],

Clarissa Glaser[5], Tobias Houska[4], Lutz Breuer[4,6], Klaus Butterbach-Bahl[1, 3], Ralf Kiese[1]

[1]Karlsruhe Institute of Technology, Institute for Meteorology and Climate Research, Atmospheric

Environmental Research (IMK-IFU), Kreuzeckbahnstrasse 19, Garmisch-Partenkirchen 82467, Germany

[2]IHE-Delft Institute for Water Education, Westvest 7 2611 AX Delft the Netherlands

[3]Pioneer Center Land-CRAFT, Department of Agroecology, University of Aarhus, Denmark

[4]Institute for Landscape Ecology and Resources Management (ILR), Research Centre for BioSystems, land use /

land cover and Nutrition (iFZ), Justus Liebig University Giessen, Giessen, 35392, Germany

[5]Center for Applied Geoscience, University of Tübingen, Tübingen, Germany

[6]Centre for International Development and Environmental Research (ZEU), Justus Liebig University Giessen,

Senckenbergstrasse 3, 35390 Giessen, Germany

*Correspondence to* Ralf Kiese (ralf.kiese@kit.edu)

## Abstract

Anthropogenic activities increase the contributions of inland waters to global greenhouse gas (GHG; $CO_2$, $CH_4$, and $N_2O$) budgets, yet the mechanisms driving these increases are still not well constrained. In this study, we quantified year-long GHG concentrations and fluxes, as well as water physico-chemical variables from 23 streams, 3 ditches, and 2 wastewater inflow sites across five headwater catchments in Germany contrasted by land use. Using mixed-effects models, we determined the overall impact of land use and seasonality on the intra-annual variabilities of these parameters. We found that land use was more significant than seasonality in controlling the intra-annual variability of GHG concentrations and fluxes. Agricultural land use and wastewater inflows in settlement areas resulted in up to 10 times higher daily riverine $CO_2$, $CH_4$, and $N_2O$ emissions than forested areas, as substrate inputs by these sources appeared to favor *in situ* GHG production processes. Dissolved GHG inputs directly from agricultural runoff and waste-water inputs also contributed substantially to the annual emissions from these sites. Drainage ditches were hotspots for $CO_2$ and $CH_4$ fluxes due to high dissolved organic matter concentrations, which appeared to favor *in situ* production via respiration and methanogensis. Overall, the annual emission from anthropogenic-influenced streams in $CO_2$-equivalents was up to 20 times higher (~71 kg $CO_2$ m$^{-2}$ yr$^{-1}$) than from natural streams (~3 kg $CO_2$ m$^{-2}$ yr$^{-1}$). Future studies aiming to estimate the contribution of lotic ecosystems to GHG emissions should therefore focus on anthropogenically perturbed streams, as their GHG emission are much more variable in space and time.



35    **Graphical abstract**

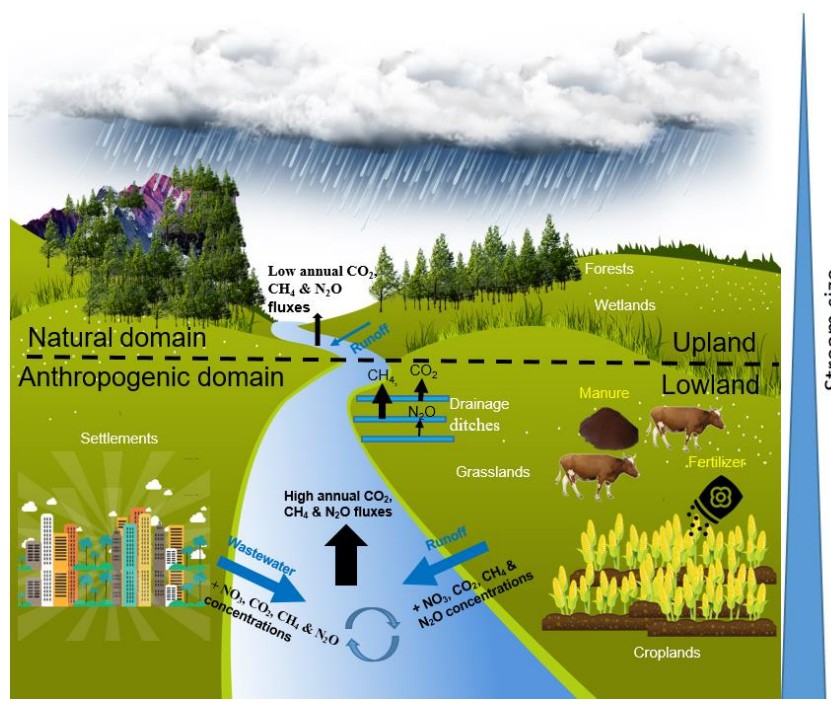





## 1 Introduction

Streams and rivers cover only a small fraction of the earth's land surface (Allen et al., 2018), yet they are significant contributors to global greenhouse gases ($CO_2$, $CH_4$, and $N_2O$), emitting approximately 7.6 (6.1–9.1) Pg-$CO_2$ equivalent into the atmosphere per year. (Li et al., 2021). Several biogeochemical process are responsible for GHG production and consumption within fluvial ecosystems. $CO_2$ production is attributed to respiration of organic matter (Battin et al., 2008). Production of $CH_4$ occurs through methanogenesis, with carbon dioxide and acetic acid as substrates under anaerobic conditions (Stanley et al., 2016). Consumption of methane is also possible through methanotrophy in oxygen rich stream waters, producing $CO_2$ in the process (Shelley et al., 2014). $N_2O$ is mainly a byproduct in nitrification (under aerobic conditions) or an intermediate product in denitrification (under anaerobic conditions), but it can also be reduced to $N_2$ in organic-rich and nitrate-poor ecosystems (Quick et al., 2019).

Anthropogenic practices such as fertilizer application and construction of drainage ditches to allow agricultural use of former wetlands alter the rates of these processes, thereby influencing in-stream GHG dynamics (Peacock et al., 2021; Wallin et al., 2020; Mwanake et al., 2019). Elevated inorganic nitrogen in streams within fertilized croplands has been shown to favor in situ $N_2O$ (e.g., Beaulieu et al., 2009), $CO_2$ production (e.g., Bodmer et al., 2016; Borges et al., 2018), and $CH_4$ production (e.g., Mwanake et al., 2022). While such trends in agricultural streams show similarities across different catchment locations, GHG emissions from streams in predominantly forested catchments with minor influences from croplands and wetlands show more diverse patterns. Some studies indicated that forest streams are hotspots for GHG fluxes (e.g., Wallin et al., 2018; Audet et al., 2019; Herreid et al., 2021), while others found the opposite with much lower fluxes in forests as compare to other land uses (e.g., Bodmer et al., 2016; Mwanake et al., 2022). Drainage ditches, which are characterized by short water residence times, high organic loads, and highly variable $O_2$ levels, can simultaneously support both aerobic and anaerobic organic carbon mineralization, driving vigorous $CH_4$ and $CO_2$ production and subsequent fluxes. In a recent meta-analysis, ditches and canals accounted for up to 3% of the global anthropogenic $CH_4$ emissions (Peacock et al., 2021). Yet, studies on them are scarce, and thus the main factors making them hotspots of carbon fluxes are still not well-constrained.

In fluvial ecosystems located in settlement areas, inflows of wastewater effluents also act as important drivers of GHG fluxes, by indirectly influencing insitu substrate availability for GHG production and through direct inflows of dissolved GHGs (e.g., Marescaux et al., 2018; Zhang et al., 2021; Wang et al., 2022). For example, in a study of urban-impacted rivers in the Seine basin in France, Marescaux et al. (2018) found elevated $CO_2$, $CH_4$, and $N_2O$ concentrations and fluxes downstream of wastewater inflows, which dispropotenately contributed higher basin-wide annual GHG fluxes. Similar findings were also found in urban-impacted rivers in China, were GHG emissions were up to 14 times higher than from other land uses (Zhang et al., 2021). Yet, studies on GHG emissions from urban-impacted fluvial ecosystems are still scarce, and therefore their contributions to riverine annual GHG budgets are not well contrained. Moreover, little is known about the interactive effects of land use and wastewater effluent inflows on riverine GHG fluxes, and whether land use is the overarching controlling factor.





Under temperate climatic conditions, pronounced seasonality regulates the availability of nutrients and
to some extent the $O_2$ in lotic ecosystems, which are both key factors driving *instream* GHG production and gas
exchange rates (Borges et al., 2018; Rocher-Ros et al., 2019; Herreid et al., 2021; Aho et al., 2022). Cold winter
periods are generally characterized by low *instream* carbon and nitrogen processing, which results in nutrient
accumulation (e.g., Herreid et al., 2021), while high *instream* C and N processing are characteristic of warm
summer periods (e.g., Borges et al., 2018; Aho et al., 2021, 2022). Seasonality in precipitation regulates
discharge, whereby heavy precipitation events or snowmelt during spring result in high discharge events. At the
same time, dry summers and winter periods are often characterized by lower discharge (e.g., Aho et al., 2022).
Discharge in turn determines the water residence times in streams, thereby influencing rates of carbon and
nitrogen processing (e.g., Borges et al., 2018; Mwanake et al., 2022). High discharge events may also increase
dissolved GHG supply from upstream terrestrial sources and *instream* GHG production depending on the
surrounding land use. For example, studies have found that during high discharge periods, streams draining
wetlands show peak $CO_2$ and $CH_4$ concentrations (e.g., Aho et al., 2019; Borges et al., 2019) and pronounced
$N_2O$ concentrations are found in streams of cropland dominated catchments (e.g., Mwanake et al., 2022).
The aforementioned interactions between seasonality and land use indicate that temporally sporadic
measurements of GHG concentrations and fluxes are limited in revealing intra-annual variations, which are
necessary for better estimating annual emissions. Yet, only a handful of studies in temperate streams have
assessed the seasonal dynamics of GHG fluxes at sampling points with contrasting land uses (e.g., Marescaux et
al., 2018; Borges et al., 2018; Herreid et al., 2021; Galantini et al., 2021). As climate change drives more
extreme discharge conditions, and as agricultural intensification and settlement areas continue to increase
(Winkler et al., 2021), studies that cover a wide array of land uses, discharge, and temperature conditions are
needed to constrain better the effects of land use in controlling intra-annual GHG flux variabilities and to unravel
synergistic or antagonistic relationships amongst them.
The main objective of this study was to assess the seasonality-land use relationships of water physico-
chemical variables and GHG concentration and fluxes by comparing temperate lotic ecosystems of forests and
wetlands with those from more human-influenced agricultural and settlement catchments. To do so, we
conducted at least tri-weekly measurements covering a full year of observations and mainly focused on
headwater streams (stream orders 1–6), which are known hotspots of fluvial emissions, but remain currently
underrepresented in global GHG datasets (Drake et al., 2018; Li et al., 2021). We hypothesize that catchment
land use is the most important control for stream GHG concentration and fluxes, with higher seasonal variability
in human-influenced ecosystems than in natural ones. Moreover, we hypothesized that drainage ditches and
headwater streams with wastewater inflow within agricultural and settlement areas are hotspots for GHG
emissions, driven by direct dissolved GHG inputs or substrate inputs that favor *in situ* GHG production.
**2    Materials and methods**
**2.1    Study areas and sampling design**
Five headwater catchments in central (Schwingbach), southeast (Loisach), and southwest (Ammer,
Goldersbach, and Steinlach) Germany were investigated in this study. The catchments covered a wide range of



fluvial ecosystems with different stream orders and land use characteristics (Table 1; Fig. 1). The catchment
boundaries for each site were determined based on the most downstream sampling location within each
catchment (Fig. 1). Elevation of the Schwingbach catchment (54 km²), located in the central-German state of
Hessen, ranges from 176–480 m above sea level (a.s.l). The catchment has a mixed land use of ~41 % mixed
forests, 46% croplands, 8 % settlement areas, and 5 % pasturelands (Wangari et al., 2022) (Fig. 1A). The climate
is warm and temperate (Cfb, Köppen climate classification), with an annual rainfall of 742 mm (monthly mean
min: 51 mm, monthly mean max: 72 mm) (1999–2019) and a mean annual temperature of 9.8 °C (monthly mean
min: 1.3 °C, monthly mean max: 18.8 °C) (1991–2021) (Climate-data.org, https://en.climate-
data.org/europe/germany/hesse/giessen-151/).

The Upper Loisach catchment (467 km², outlet Eschenlohe town) is located in the mountainous region
of the Bavarian Alps, Germany. The catchment is characterized by a pronounced relief and steep slopes, with
elevations ranging from 616–2,963 m a.s.l. Land use in the catchment comprises coniferous and deciduous
forests interspersed with natural grasslands and rocky surfaces on the mountain slopes (78%). At the valley
bottom, the land use is mainly settlement areas (9%), managed grasslands (8%), and wetlands (5%) (Fig. 1B).
The climate is cold and temperate (Dfb, Köppen climate classification), with annual precipitation of 1,693 mm
(monthly mean min: 87 mm, monthly mean max: 207 mm) (1999–2019) and mean annual temperature of 3.8 °C
(monthly mean min: -6.6 °C, monthly mean max: 13.1 °C) (1991–2021) (Climate-data.org, https://en.climate-
data.org/europe/germany/free-state-of-bavaria/garmisch-partenkirchen-8762/).

The other three catchments are sub-catchments of the Neckar river (Fig. 1C). The Goldersbach (116
km²), a tributary of the main Ammer stream, is a forested catchment (95%), with elevations ranging from 366–
583 m a.s.l. The Steinlach catchment (513 km²) is also dominated by forests (74%), with agricultural lands
(croplands and grasslands) and settlement areas occupying 21% and 5% of the landscape, respectively. The
elevation range of the hilly area is 321–878 m a.s.l (Fig. 1C). The Ammer catchment (304 km², outlet
Pfäffingen) is dominated by agricultural lands (80%), with 11% forests and 9% settlement areas (Fig. 1C). It has
moderate slopes with an elevation ranging from 319–610 m a.s.l. The Ammer stream is a gaining stream fed by
an extensive groundwater karst system and has significant discharge levels even during the driest periods of the
year (Glaser et al., 2020). The climate is warm and temperate (Cfb, Köppen climate classification), with a mean
annual rainfall of 923 mm (monthly mean min: 63 mm, monthly mean max: 98 mm) (1999–2019) and a mean
annual temperature of 9.3 °C (monthly mean min: 0.2 °C, monthly mean max: 18.6 °C) (1991–2021) (Climate-
data.org, https://en.climate-data.org/europe/germany/baden-wuerttemberg/tuebingen-22712/).

Across the five catchments, a total of 28 sites at headwater streams (N=23, orders 1–6, defined after
Strahler, 1952), drainage ditches (N=3) and waste water outflows (N=2, Text A1) were sampled every 2–3 weeks
for an entire year (Table 1, Fig. 1). The Schwingbach and Loisach catchments were sampled from June 2020 to
June 2021 while the Goldersbach, Ammer, and Steinlach catchments, were sampled from April 2021 to April
2022.






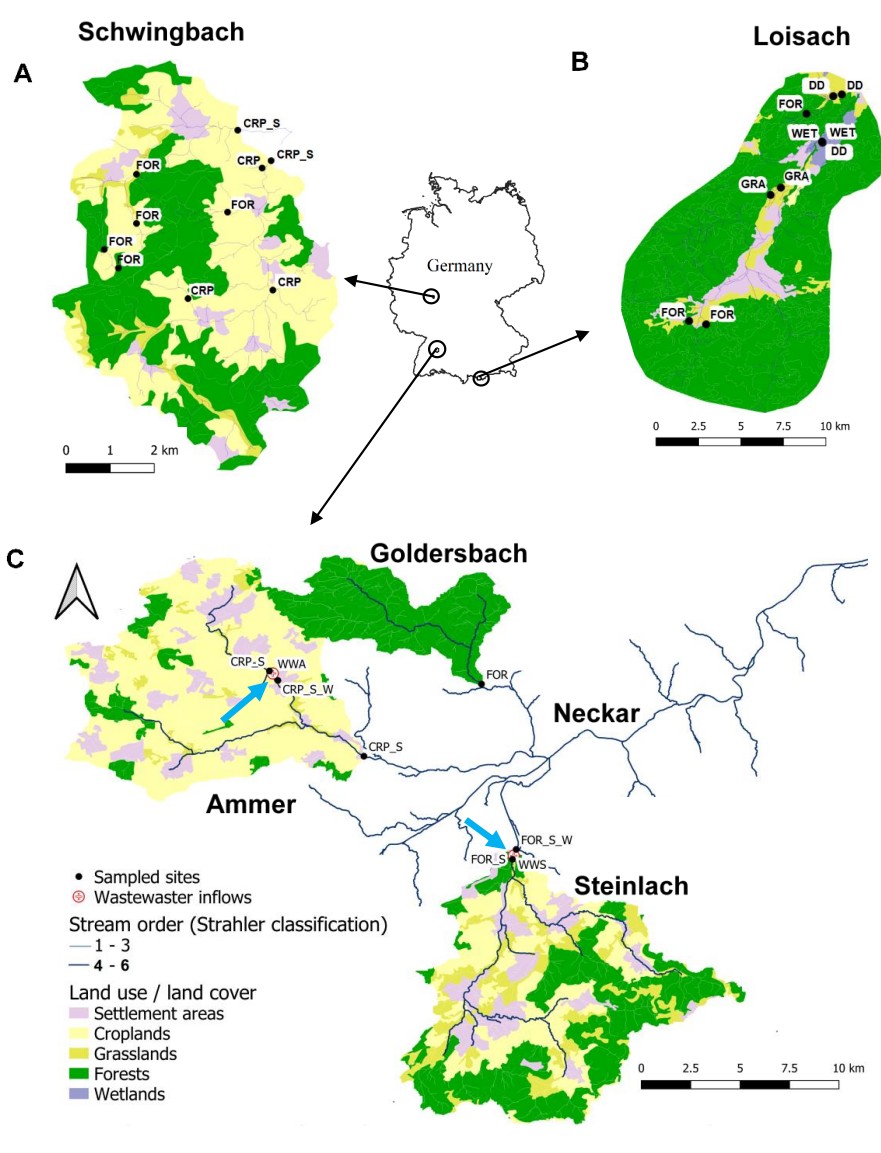



Fig. 1: Land cover maps of the (A) Schwingbach, (B) Loisach, and (C) Neckar sub-catchments (Goldersbach,

Ammer, and Steinlach) derived from the Corine Land Cover 2018 inventory with a 25 ha spatial resolution

(https://land.copernicus.eu/pan-european/corine-land-cover/clc2018?tab=mapview). Black dots with labels

(abbreviations explained in Table 1) represent sampled headwater streams and drainage ditch sampling points.

Wastewater inflows sampled are indicated by blue arrows on the maps. Drainage ditches in Loisach catchment

were dug in the 1930s to 1960s to lower water levels to improve grassland productivity in areas formerly

occupied by wetlands.



**2.2    Sub-catchment delineation and land use classification**

Sub-catchments for each sampling point in the Loisach, Goldersbach, Steinlach, Ammer and

Schwingbach catchments were delineated in QGIS from a Digital Elevation Model (DEM) (EU-DEM v1.1) with
a 25 m resolution (European Copernicus mission, retrieved August 1, 2021, https://land.copernicus.eu/imagery-
in-situ/eu-dem/eu-dem-v1.1). Land use/ land cover percentages of all the delineated sub-catchments were
calculated from Corine Land Cover 2018 survey with a 25 ha spatial resolution (retrieved August 1, 2021,
https://land.copernicus.eu/pan-european/corine-land-cover/clc2018?tab=mapview). For the purposes of data
analysis, we classified sub-catchments according to their dominant land cover (>50% of the total area) into forest
(FOR), cropland (CRP), grassland (GRA), and wetland (WET), and further differentiated sub-catchments with
influence of settlement areas (S) and wastewater inflows (W). (Table 1). As drainage ditches (DD) in the Loisach
catchment were also added as extra land use category, this classification resulted in a total of 9 land use classes
(for details see Table 1).
**2.3    Hydrological and water physico-chemical characteristics**

In the Loisach and Schwingbach catchments, discharge was calculated (Gore, 2007) from stream depth

and velocity measurements using an electromagnetic sensor (OTT-MF-Pro, Hydromet, Germany). For streams in
the Neckar sub-catchments, velocity was measured using the electromagnetic sensor (OTT-MF-Pro, Hydromet,
Germany), and depth and discharge were obtained directly from gauging stations maintained by the water
authority of the state of Baden-Württemberg (https://udo.lubw.baden-wuerttemberg.de/public/index.xhtml). The
slope of a ~5 m reach at each sampling point was measured using a laser rangefinder with a slope function
(Nikon Model: 8381, Japan). The slopes and velocities were used to model the site-specific gas transfer
velocities (k in m d$^{-1}$) for the quantification of daily GHG fluxes per unit stream surface area (mass m$^{-2}$ d$^{-1}$) (see
details in the flux calculation section).

Discharge measurements at each sampling location and at every sampling event were complemented by

*in situ* measurements of water temperature (°C), electrical conductivity (µS cm$^{-1}$), dissolved oxygen (DO) (mg L$^{-1}$
), and pH using the Pro DSS multiprobe (YSI Inc., USA). Water samples for nutrient and organic carbon
analyses were also collected and filtered on-site through polyethersulfone (PES) filters (0.45 µm pore size, pre-
leached with 60 ml of miliq water). The samples were stored in triplicates in 30 ml acid-washed HDPE sample
bottles and transported within 24 h to the laboratories at Karlsruhe Institute of Technology, Campus Alpin,
Justus Liebig University Giessen, or the University of Tübingen. On arrival, all samples were immediately
frozen for later analysis.

After unfreezing the samples overnight in a 4° C refrigerator, the samples were directly analyzed for

dissolved organic carbon (DOC), total dissolved nitrogen (TDN), nitrate (NO$_3$-N), and ammonium (NH$_4$-N)
concentrations. Dissolved organic nitrogen (DON) concentrations were estimated as the difference between the
TDN and dissolved inorganic nitrogen DIN (NO$_3$-N + NH$_4$-N) concentrations. DIN concentrations were
determined using colorimetric methods, and the absorbance of the samples was measured using a microplate
spectrophotometer (Model: Epoch, BioTek Inc., USA). NO$_3$-N concentrations were analyzed based on reactions
with the Griess reagent (Patton & Kryskalla, 2011), and NH$_4$-N concentrations were analyzed using the
indophenol method (Bolleter et al., 1961). The DOC concentrations were measured as non-purgeable organic



carbon (NPOC) using a TOC/ TN analyzer (Analytica-Jena; multi N/C 3100, Germany) after pre-treating the
sample with 25% HCl acid to remove the dissolved inorganic carbon (DIC). The TDN concentrations were
analyzed simultaneously with the same instrument (Analytica-Jena; multi N/C 3100, Germany).

**2.4    Gas sampling, analysis, and calculations of annual areal fluxes**

GHG samples of stream, ditch and waste water were collected in triplicates simultaneously with the

water physico-chemical samples using the headspace equilibration technique (Raymond et al., 1997). In brief, 80
ml of background water was equilibrated with 20ml of atmospheric air in a syringe at *in situ* water temperatures,
and the headspace gas sample transferred into 10ml glass vials for GHG concentration analysis in the laboratory
of the Karlsruhe Institute of Technology, Campus Alpin (see full sampling details in Mwanake et al., 2022).
Atmospheric air samples were taken twice (morning and afternoon) on each sampling day to correct for
background atmospheric GHG concentrations. GHG concentrations from the headspace were analyzed using an
SRI gas chromatograph (8610C, Germany) with an electron capture detector (ECD) for $N_2O$ and a flame
ionization detector (FID) with an upstream methanizer for simultaneous measurements of $CH_4$ and $CO_2$
concentrations. Dissolved GHG concentrations in the stream water were calculated from post-equilibration gas
concentrations in the headspace after correcting for atmospheric (ambient) GHG concentrations (e.g., Aho et al.,
2019; Mwanake et al., 2022).

Daily diffusive fluxes ($F$) (moles $m^{-2}$ $d^{-1}$) of the GHGs were estimated using Fick's Law of gas

diffusion, where the $F$ is the product of the gas exchange velocity ($k$) (m $d^{-1}$) and the difference between the
stream water ($C_{aq}$) (moles $m^{-3}$) and the ambient atmospheric gas concentration in water assuming equilibrium
with the atmosphere ($C_{sat}$) (moles $m^{-3}$) (Equation 1). GHG concentrations and fluxes were expressed in mass
units by multiplying by the respective molar masses.

$$F = k \ (C_{aq} - C_{sat}) \qquad (1)$$

The temperature-specific gas transfer velocities ($k$) for each of the gases were calculated from

normalized gas transfer velocities ($k_{600}$) (m $d^{-1}$) (corresponding to the $k$ of $CO_2$ at 20° C with a Schmidt number
of 600) and temperature-dependent Schmidt numbers (Sc) (unit-less) of the respective gases (Equation 2).

$$k = k_{600} \ \times \ \left(600/Sc\right)^{0.5} \qquad (2)$$

The $k_{600}$ was modeled using Equation 3 (drawn from equation 4 in Table 2 of Raymond et al. (2012)), which was
calibrated from headwater streams of similar characteristic of our study sites, where V is stream velocity (m $s^{-1}$)
and S is the slope (m $m^{-1}$).

$$k_{600} = VS^{0.76} \ \times \ 951.5 \qquad (3)$$

Before chosing the aforementioned equation for modeling the $k_{600}$ values, we compared the $k_{600}$ values

calculated from all seven empirical models from Raymond et al 2012. The predicted $k_{600}$ values from models
3,4,5 and 6 in Table 2 of Raymond et al. (2012), which all use velocity and slope as input parameters, were
mostly similar for the three discharge periods and across all stream orders 1–6 (ANOVA; p>0.05). In contrast,
the calculated $k_{600}$ values from equation 1, 2 and 7, which use a stream depth parameter, were higher (ANOVA;
p<0.05), particularly from the higher stream orders (5–6). This finding is inconsistent with the energy dissipation



model of turbulent streams where $k_{600}$ is predicted to decrease with stream order. We therefore interpreted this to
indicate a breakdown of these models for higher stream orders. This is also in agreement with recommendations
from Raymond et al. 2012, and we therefore choose not to use models 1, 2 and 7 for this study. Out of the
remaining equations 3, 4, 5 and 6, we used equation 4, which calculates $k_{600}$ based on the slope and velocity
parameters, and was also in line with several previous studies spanning a wide range of stream orders similar to
our study. (See, Aho et al., 2019; Borges et al., 2019; Mwanake et al., 2019; Hall & Ulseth, 2020; Aho et al.,
2021; Mwanake et al., 2022). The uncertainties in the modelled gas transfer velocities were reduced in this study
by parametrization of the velocities and slopes based on actual field measurements of both variables. Equation 3
was also used to estimate the gas transfer velocities in the drainage ditches that also had a measurable flow
velocity and slope.
Water-to-atmosphere fluxes for all three GHGs across all land use classes in each subcatchment were
calculated from the mean daily $CO_2$, $CH_4$ and $N_2O$ fluxes during different discharge conditions. Total GHG
fluxes were expressed as $CO_2$ equivalents emissions (mg $CO_2$-eq m$^{-2}$ d$^{-1}$) computed from global warming
potentials (GWP$_{100}$) using 28 for $CH_4$ and 298 for $N_2O$ (IPCC, 2014). In order to scale tri-weekly measurements
to annual flux estimates, we followed the procedure developed in Mwanake et al. (2022). Briefly, we classified
each sampling date of every location into low, medium, or high discharge conditions according to whether
normalized discharge fell in the 0–33% percentile (low), 34–66% (medium), or 67-100% (high) days.
Normalized discharge for each site was determined by dividing each absolute discharge measurement for every
site visit during the year with the maximum measured discharge. The number of days in each discharge period
were estimated as the ratio of the number of observations in each discharge period to the total number of flux
observations in individual land use classes in each catchment. $CO_2$ equivalents fluxes were then calculated for
the three different discharge periods in each land use class by multiplying the daily mean $CO_2$ equivalents flux
measured during each period and the number of days within each period. Annual fluxes were finally estimated
by summing up the emissions of the low, medium, and high discharge periods for the individual land use classes
in each catchment.



**2.5    Statistical analysis**

Linear mixed-effects models were used to investigate the effect of seasonality and land use on water
physico-chemical variables, GHG concentrations, and fluxes ("lme4" package in R version 4.1.1). Fixed effects
in the models consisted of land use classes in each catchment (Table 1) and seasons: summer June 1–August 31,
autumn September 1–November 30, winter December 1–February 28, and spring March 1–31[th] May. Random
effects accounting for repeated measures were also included in the models. Model performance was assessed
based on the distribution of residuals (i.e residuals should be normally distributed with a mean close to zero) and
conditional $r^2$ values calculated from significant models (p-value <0.05) ("MuMln" package in R). A Tukey post-
hoc test (p-value <0.05) of least-square means was used on the mixed models to identify individual differences
within each categorical fixed effect. GHG concentration and flux data and other water physico-chemical
variables were transformed using the natural logarithm to meet the assumption of normality. Because we
quantified occasional negative fluxes in some of our sites, constant flux values of 50 mg m$^{-2}$ d$^{-1}$ for $CO_2$-C, 0.5
mg m$^{-2}$ d$^{-1}$ for $CH_4$-C, and 10 µg m$^{-2}$ d$^{-1}$ for $N_2O$-N were added to the fluxes to enable the natural logarithm
transformations.

Path analysis from structural equation models (SEMs, "lavaan" package in R version 4.1.1) were used to
determine how environmental factors linked to seasonality and landuse, directly or indirectly influenced *in*
*stream* GHG production and consumption processes as well as external GHG sources, i.e., dissolved GHGs
inputs to the streams originating from either wastewater inflows or terrestrial landscapes which were not
produced *in situ*. In brief, these SEMs were constructed on the basis of causal relationships between exogenous
variables (interpreted as ultimate drivers of GHG concentrations) and endogenous variables, which are affected
by the exogenous variables and also act as immediate drivers that affect GHG concentrations. Endogenous
variables in the models, which are known to influence *in situ* biogeochemical GHG production and consumption
processes directly, included dissolved oxygen DO (% saturation), DOC (mg L$^{-1}$), $NH_4$-N (mg L$^{-1}$), and $NO_3$-N
(mg L$^{-1}$) concentrations (Battin et al., 2008; Stanley et al., 2016; Quick et al., 2019). The exogenous variables in
the models, which influence *in situ* GHG concentrations either directly by facilitating dissolved GHG inputs or
indirectly by controlling the endogenous variables, were water temperature (°C) (a proxy for different seasons),
stream velocity V (m s$^{-1}$), % upstream agricultural area for each sampling point (AGR: grassland + cropland
area) and wastewater inflows (WW:  Boolean numbers, i.e., 1 for the presence of wastewater inflow and 0 for
absence).

The hypothesized relationships between the endogenous and exogenous drivers of instream GHG
concentrations were assessed in the overall theoretical SEM, which is made up of several multivariate regression
equations shown in Equations 4-8. To get the best-fit SEM, removal of parts of the theoretical SEM was done
manually until the model with the highest parsimony fit index (PNFI), and a root mean squared error of
approximation (RMSEA) of <=0.05 was found (Schumacker and Lomax, 2016). Graphical representations of the
significant relationship pathways from the best-fit model, including standardized slope parameter estimates, were
done using the "semPlot" package in R software.

$Log_e\ GHG\ concentration = DO + DOC + stream\ velocity + water\ temperature + Log_eNO_3 +$
$Log_e\ NH_4 + wastewater\ inflow\ + agricultural\ area$
(4)



$$Log_e NO_3 = DO + Log_e\ NH_4 + DOC\ + wastewater\ inflow\ + agricultural\ area\ +$$
$stream\ velocity$ (5)
$$Log_e\ NH_4 = DO + DOC + wastewater\ inflow\ + agricultural\ area\ +$$
$stream\ velocity$ (6)
$$DOC = \ wastewater\ inflow\ + agricultural\ area\ + stream\ velocity$$ (7)
$$DO = \ DOC + \ wastewater\ inflow\ + agricultural\ area\ + stream\ velocity$$ (8)





Table 1: Summary descriptions of sampling sites located in the Schwingbach, Loisach, and Neckar sub-catchments (Goldersbach, Ammer and Steinlach)
(Fig. 1). The land use (%) was calculated for the site-specific upstream sub-catchments based on the Corine Land Cover 2018 survey of Europe (See main text for details).

| Main Catchment | Site | Stream order | Coordinates (decimal degrees) | | Sub-catchment area (Ha) | Elevation at sampling point | Sub-catchment Landuse / landcover (%) | | | | | Wastewater inflow | Main sub-catchment landuse class | Main landuse Abreviations |
|---|---|---|---|---|---|---|---|---|---|---|---|---|---|---|
| | | | Latitude | Longitude | | | Forest | Wetland | Grassland | Cropland | Urban | | | |
| Loisach | Stream | 1 | 47.5694 | 11.1554 | 4 | 651 | 40 | 60 | 0 | 0 | 0 | No | Wetland | WET |
| Loisach | Stream | 2 | 47.5689 | 11.1556 | 10 | 645 | 22 | 78 | 0 | 0 | 0 | No | Wetland | WET |
| Loisach | Stream | 1 | 47.5440 | 11.1193 | 11 | 660 | 0 | 0 | 100 | 0 | 0 | No | Grassland | GRA |
| Loisach | Stream | 1 | 47.5399 | 11.1105 | 13 | 663 | 19 | 0 | 81 | 0 | 0 | No | Grassland | GRA |
| Loisach | Stream | 1 | 47.4670 | 11.0537 | 40 | 750 | 86 | 0 | 14 | 0 | 0 | No | Forest | FOR |
| Loisach | Stream | 2 | 47.4691 | 11.0394 | 75 | 756 | 99 | 0 | 0 | 0 | 1 | No | Forest | FOR |
| Loisach | Stream | 2 | 47.5858 | 11.1429 | 102 | 719 | 100 | 0 | 0 | 0 | 0 | No | Forest | FOR |
| Loisach | Drainage ditch | | 47.5963 | 11.1730 | 11 | 630 | 27 | 0 | 73 | 0 | 0 | No | Drainage ditch | DD |
| Loisach | Drainage ditch | | 47.5953 | 11.1657 | 11 | 645 | 43 | 57 | 0 | 0 | 0 | No | Drainage ditch | DD |
| Loisach | Drainage ditch | | 47.5696 | 11.1550 | 17 | 630 | 47 | 0 | 53 | 0 | 0 | No | Drainage ditch | DD |
| Schwingbach | Stream | 1 | 50.5051 | 8.6127 | 41 | 297 | 96 | 0 | 0 | 4 | 0 | No | Forest | FOR |
| Schwingbach | Stream | 1 | 50.4695 | 8.6179 | 60 | 187 | 0 | 0 | 0 | 100 | 0 | No | Cropland | CRP |
| Schwingbach | Stream | 2 | 50.4811 | 8.5407 | 62 | 241 | 98 | 0 | 2 | 0 | 0 | No | Forest | FOR |
| Schwingbach | Stream | 1 | 50.4756 | 8.5472 | 67 | 334 | 86 | 0 | 0 | 14 | 0 | No | Forest | FOR |
| Schwingbach | Stream | 2 | 50.4922 | 8.5971 | 220 | 260 | 47 | 0 | 0 | 53 | 0 | No | Cropland | CRP |
| Schwingbach | Stream | 2 | 50.5032 | 8.5553 | 220 | 272 | 65 | 0 | 0 | 35 | 0 | No | Forest | FOR |
| Schwingbach | Stream | 2 | 50.4887 | 8.5555 | 268 | 204 | 83 | 0 | 0 | 17 | 0 | No | Forest | FOR |
| Schwingbach | Stream | 1 | 50.4669 | 8.5792 | 355 | 207 | 14 | 0 | 0 | 84 | 2 | No | Cropland | CRP |
| Schwingbach | Stream | 3 | 50.5050 | 8.6148 | 2337 | 183 | 37 | 0 | 6 | 48 | 9 | No | Cropland+settlement | CRP_S |
| Schwingbach | Stream | 3 | 50.5166 | 8.5992 | 5345 | 189 | 44 | 0 | 4 | 45 | 7 | No | Cropland+settlement | CRP_S |
| Goldersbach (Neckar) | Stream | 5 | 48.5588 | 9.0591 | 11623 | 367 | 97 | 0 | 0 | 3 | 0 | No | Forest | FOR |
| Ammer (Neckar) | Stream | 5 | 48.5649 | 8.8986 | 26157 | 379 | 11 | 0 | 1 | 84 | 4 | No | Cropland+settlement | CRP_S |
| Ammer (Neckar) | Stream | 6 | 48.5640 | 8.8997 | 26361 | 377 | 11 | 0 | 1 | 83 | 5 | Yes | Cropland+settlement+wastewater | CRP_S_W |
| Ammer (Neckar) | Stream | 6 | 48.5271 | 8.9615 | 30441 | 348 | 14 | 0 | 2 | 77 | 8 | No | Cropland+settlement | CRP_S |
| Steinlach/Neckar | Stream | 6 | 48.4796 | 9.0634 | 51332 | 348 | 74 | 0 | 10 | 11 | 4 | No | Forest+settlement | FOR_S |
| Steinlach/Neckar | Stream | 6 | 48.4812 | 9.0639 | 51332 | 344 | 74 | 0 | 10 | 11 | 4 | Yes | Forest+settlement+wastewater | FOR_S_W |
| Ammer/Neckar | Wastewater effluent | | 48.5644 | 8.8993 | | | | | | | | | Wastewater | WWA |
| Steinlach/Neckar | Wastewater effluent | | 48.4805 | 9.0636 | | | | | | | | | Wastewater | WWS |




## 3 Results

### 3.1 Hydrological variables

Across all sampling points and seasons, tri-weekly sampled stream velocity measurements (annual mean ± SE) were two-folds higher for streams ($0.19 \pm 0.009$ m s$^{-1}$, range: 0.01- 1.17) than ditches ($0.05 \pm 0.06$ m s$^{-1}$, range: 0.01–0.23) (Fig A1). Seasonality had an overall significant effect on stream velocities across all sampling points, with higher stream velocities observed in spring ($0.24 \pm 0.02$ m s$^{-1}$) than in autumn ($0.12 \pm 0.01$ m s$^{-1}$) (Table 2; Table B2). Discharge in streams (3.9–18,500 L s$^{-1}$) and in ditches (0.1–37 L s$^{-1}$) was highly variable, reflecting differing stream sizes and seasonal variability (Fig. A1). The Neckar sub-catchments, dominated by streams (orders 5 - 6 ), had an order of magnitude higher mean annual discharge ($874.7 \pm 178$ L s$^{-1}$) than the streams in the other catchments (Loisach: $50.5 \pm 6$ L s$^{-1}$ and Schwingbach: $26.7 \pm 4$ L s$^{-1}$). The average discharge at the stream  and ditch sampling points in all our study catchments were 3 to 5-fold higher in spring and summer ($384.1 \pm 96$ and $526.4 \pm 171$ L s$^{-1}$, respectively) than in autumn and winter ($86.25 \pm 13.07$ and $157.3 \pm 31.58$, respectively; Table 2; Table B2).



Table 2: Results of multiple linear mixed-effects models predicting the effect of seasonality (summer, autumn,
winter, and spring) and sub-catchment land use (Table 1) on stream velocity, discharge, water physico-chemical
variables, GHG concentration, gas-transfer velocity, and GHG flux. The model performance was assessed based
on conditional $r^2$ and the distribution of residuals, including the variances explained by fixed effects and repeated
mea sures' random effects.

| | | Type 2 ANOVA table | |
| --- | --- | --- | --- |
| | | Season (df=3) | Land use (df=11) |
| **Dependent variables** | **Conditional $r^2$** | **F-statistic/significance** | **F-statistic/significance** |
| **Water physico-chemical and hydrological variables** | | | |
| Temperature (° C) | 0.87 | 66.3*** | 9.1*** |
| pH | 0.80 | 3.1* | 97.8*** |
| DO (mg L$^{-1}$) | 0.83 | 20.1*** | 143.7*** |
| Electrical Conductivity (µs cm$^{-1}$) | 0.83 | 4.9** | 86.1*** |
| NO$_3$-N (mg L$^{-1}$) [a] | 0.80 | 4.9** | 141*** |
| NH$_4$-N (mg L$^{-1}$) [a] | 0.60 | ns | 32.3*** |
| TDN (mg L$^{-1}$) [a] | 0.79 | 5.6** | 93.8*** |
| DON (mg L$^{-1}$) [a] | 0.55 | ns | 13.9*** |
| DOC (mg L$^{-1}$) [a] | 0.59 | ns | 47.3*** |
| DOC:DIN | 0.84 | 3.2* | 133.2*** |
| DOC:DON | 0.63 | ns | 15.1*** |
| Velocity [a] | 0.59 | 3.7* | 34.5*** |
| Discharge [a] | 0.86 | 4.6** | 96.9*** |
| **$k_{600}$, Gas concentration and flux** | | | |
| CO$_2$-C concentration (µg L$^{-1}$) [a] | 0.86 | 25.6*** | 219.3*** |
| CH$_4$-C concentration (µg L$^{-1}$) [a] | 0.89 | ns | 273.1*** |
| N$_2$O-N concentration (ng L$^{-1}$) [a] | 0.75 | 3.3* | 69*** |
| $k_{600}$ (m d$^{-1}$) [a] | 0.57 | ns | 31.2 *** |
| CO$_2$-C flux (mg m$^{-2}$ d$^{-1}$) [a] | 0.57 | ns | 50.2*** |
| CH$_4$-C flux (mg m$^{-2}$ d$^{-1}$) [a] | 0.79 | ns | 113*** |
| N$_2$O-N flux (µg m$^{-2}$ d$^{-1}$) [a] | 0.70 | 3.9* | 75.6*** |
| Total fluxes CO$_2$-eq (g m$^{-2}$ d$^{-1}$) [a] | 0.67 | ns | 67*** |
| Level of significance (p-value) | [a] Natural logarithim transformation | | |
| * <0.05 | | | |
| ** <0.01 | | | |
| *** <0.001 | Conditional $r^2$ = Variance explained by fixed and random effects of sampling date | | |
| ns >0.05 | df= degrees of freedom | | |




## 3.2 Water physico-chemical variables

### 3.2.1 Seasonal variation

Water temperature, DO, and pH ranged from 0.9–24° C, 1.1–15.7 mg O$_2$ L$^{-1}$, and 6.7–9.0, respectively.
Streams in the mountainous Loisach catchment had a mean annual (± SE) water temperature of 9.0 ± 0.2 °C,
which was ~1 °C colder than streams of the Schwingbach catchment (10.0 ± 0.4 °C) and 3 degrees colder than



streams in the Neckar sub-catchments ($12.0 \pm 0.3$ °C). The annual ranges of $NH_4$-N, $NO_3$-N, DON, TDN, and
DOC concentrations across all catchments were 0.05– 1.0 mg L$^{-1}$, 0.5–14.8 mg L$^{-1}$, 0.05–10.9 mg L$^{-1}$, 0.6–17.0
mg L$^{-1}$, and 0.9–16.0 mg C L$^{-1}$, respectively. DO, $NO_3$, and TDN concentrations showed significant seasonal
variability (Table 2, Table B2). DO was higher in winter and spring than in summer and autumn. $NO_3$-N and
TDN concentrations were highest in winter and lowest in autumn and summer, while $NH_4$-N, DOC and DON
showed no significant seasonal variation (Table 2; Table B2). We additionaly calculated DOC:DIN and
DOC:DON molar ratios, which had interquartile ranges from 0.9–4.9 and 4.1–29.0, respectively. DOC:DIN
ratios showed significant seasonal variability, with higher values in summer and spring than in winter, while no
seasonal variability was found for DOC:DON ratios (Table 2: Table B2).

### 3.2.2    Land use variation

Catchment land use was more significant than seasonality in explaining the variability of most water
physico-chemical variables (Table 2). In the Loisach catchment, ditches had up to 2.6 times lower DO and up to
8 times lower $NO_3$-N concentrations than the streams across all land use types (Fig. 2; Table B3). In contrast,
$NH_4$-N and DOC concentrations, as well as the DOC:DIN ratio were 6-10 times higher in the ditches than in the
streams (Fig. 2; Table B3). In the Neckar sub-catchments, forested streams had 1-2 times higher DO and DOC
concentrations than cropland, settlement, and wastewater-influenced streams. The opposite was true for $NO_3$-N
and DON concentrations, which were an order of magnitude higher in the cropland, settlement, and wastewater-
influenced streams than in the forested streams (Fig. 2; Table B3). As a result, DOC:DIN and DOC:DON ratios
in the Neckar sub-catchments were therefore higher in forested streams than in cropland, settlement, and
wastewater-influenced streams (Table B3).
In addition, cropland streams directly receiving wastewater inflows also had significantly lower DO and
higher DOC than cropland streams without wastewater inflows (Fig. 2; Table B3). While $NO_3$-N and DON
concentrations were not significantly different in cropland streams with or without wastewater inflows, the
concentrations of both variables was slightly higher in cropland streams with wastewater inflows (Table B3). In
streams of the Schwingbach catchment, surrounding croplands and settlement areas also influenced $NO_3$-N
concentrations, which were up to 3-fold higher than in the forested streams. Across all the three catchments, DO
concentrations, DOC:DIN and DOC:DON ratios were higher in the forested streams and decreased in streams of
sub-catchments with predominant agricultural land uses or settlement areas, while the opposite was found for
$NO_3$-N and DON concentrations (Table B3). Additionally, forested streams in the Loisach catchment had an
order of magnitude higher DOC:DON ratios than forested streams in the Neckar and Schwingbach catchments
(Table B3).




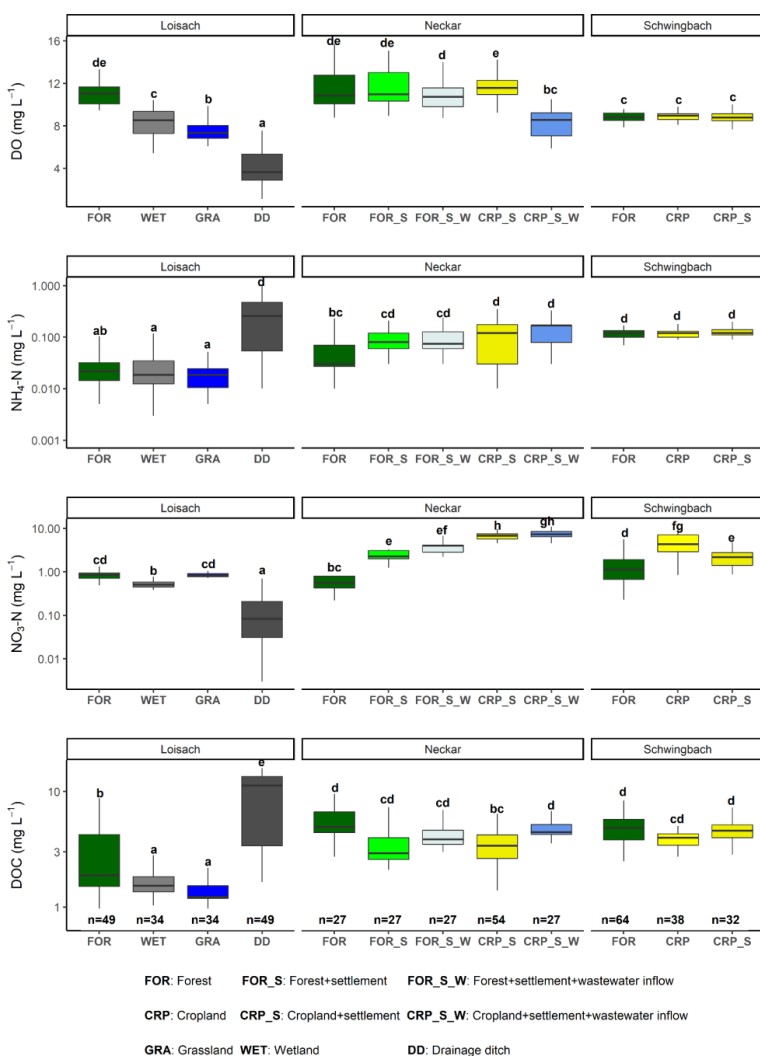

Fig. 2: Boxplots of DO, $NH_4$-N, $NO_3$-N, and DOC concentrations in stream and ditch waters in the three catchments grouped by dominating land uses (see Table 1 methods). Letters on top of the boxplots represent significant differences ($p<0.05$) among land use classes across the three catchments based on Tukey post-hoc analyses from the linear mixed-effects model results (Table 2).

## 3.3 GHG concentrations and fluxes

### 3.3.1 Seasonal variation

In all headwater streams, $CH_4$ and $N_2O$ concentrations varied greatly, spanning three orders of magnitude, i.e., from 0.03– 58 µg-C $L^{-1}$ ($pCH_4$ 1.3–2,145 µatm) for $CH_4$ and from 20–18,717 ng-N $L^{-1}$ ($pN_2O$



21– 15,813 natm) for $N_2O$. In contrast, $CO_2$ concentrations varied less, spanning only one order of magnitude
from 219–4,868 µg-C $L^{-1}$ ($pCO_2$ 369–7,979 µatm). GHG concentrations in ditches also varied widely, with $CH_4$,
$N_2O$ and $CO_2$ concentrations spanning 1-2 orders of magnitude ranging from 27–831 µg-C $L^{-1}$ ($pCH_4$ 1,469–
34,482 µatm), 56–1,540 ng-N $L^{-1}$ ($pN_2O$ 35–1,512 natm), and 1,722– 9,746 µg-C $L^{-1}$ ($pCO_2$ 2,888–13,400
µatm), respectively (Fig. A2–A5).

Streams and drainage ditches across all seasons were predominantly sources of atmospheric $CH_4$, $N_2O$,
and $CO_2$, as indicated by concentrations mostly above atmospheric background and the positive flux values
displayed in Figure 3. $CO_2$ fluxes from streams ranged from -0.05–179 g C $m^{-2}$ $d^{-1}$ (mean 19 g C $m^{-2}$ $d^{-1}$), $CH_4$
fluxes ranged from -0.40–325 mg C $m^{-2}$ $d^{-1}$ (mean 30 mg C $m^{-2}$ $d^{-1}$), and $N_2O$ fluxes ranged from -9.2–199.5 mg
N $m^{-2}$ $d^{-1}$ (mean 12 mg N $m^{-2}$ $d^{-1}$). $CO_2$ and $CH_4$ fluxes from the ditches varied between 2–63 g C $m^{-2}$ $d^{-1}$ (mean
13.7 g C $m^{-2}$ $d^{-1}$) and from 117–7,933 mg C $m^{-2}$ $d^{-1}$ (mean 1,532 mg C $m^{-2}$ $d^{-1}$), respectively, while $N_2O$ fluxes
ranged from -0.8–7.1 mg N $m^{-2}$ $d^{-1}$ (mean 1.2 mg N $m^{-2}$ $d^{-1}$).

Seasonal variation in GHG concentrations and fluxes were GHG dependent and varied across the
different land uses within each catchment (Fig. 3; Fig. A2, A3, and A4). In the Loisach catchment, there was a
decline in *instream* $CO_2$ concentrations in the summer followed by a subsequent increase in autumn, particularly
at non-forested sampling points (Fig. A2). Similar *instream* $CO_2$ concentration trends with lower values in the
summer season and increasing values in autumn, were also found for non-forested streams of the Neckar sub-
catchments (Fig. A3). However, non-forested streams of the Schwingbach catchments showed slightly different
trends, with a decline of $CO_2$ concentrations in spring and an increase of $CO_2$ concentrations in the late summer
season. (Fig. A4). Considering all data over all catchments, seasonality had an overall significant effect (p<0.05)
with $CO_2$ concentrations in summer being 1.6 times lower than in autumn, while $CO_2$ fluxes showed no
significant seasonal variability (Table 2; Table B2).

In contrast to $CO_2$, $N_2O$ concentrations in the Loisach and Schwingbach catchments decreased from
summer to autumn but increased again towards the beginning of the winter season (Fig. A2, A4). In autumn,
$N_2O$ concentrations at first and second order forested streams in the Loisach and Schwingbach catchments were
often below atmospheric concentrations (Fig. A2, A4), characterizing these sites as $N_2O$ sinks (Fig. 2). A similar
autumn decline in $N_2O$ concentrations was not observed in the streams of the Neckar sub-catchments, but rather,
$N_2O$ concentrations increased from autumn to winter (Fig. A3). Across all catchments and sampling points, $N_2O$
concentrations were 2.4 times higher in winter than in the other seasons (Table B2). $N_2O$ fluxes were up to 1.6
times higher in summer and winter than in autumn and spring (Fig. 3; Table B2), which represented periods of
either high $N_2O$ concentrations and moderate gas transfer velocities (winter) or moderate $N_2O$ concentrations and
high gas transfer velocities (summer) (Table B2).

$CH_4$ concentrations showed a seasonal pattern only in the Schwingbach catchment (Fig. A4), which
showed a decline from summer through autumn and winter. This trend was not observed for the other
catchments (Fig. A2, A3) and resulted in non-significant seasonal effect on both concentrations and fluxes when
all data from all catchments were considered together (Table 2; Table B2). Overall, strong seasonal trends of
GHG fluxes throughout the year were mostly found in human-influenced land use classes such as streams and



ditches in grasslands, croplands, and settlement areas, but not at streams whose sub-catchments were dominated
by forests or wetlands (Fig. 3).



Fig. 3: Monthly mean ± SE of $CO_2$, $CH_4$, and $N_2O$ fluxes across all 26 sampled streams and ditches in    the Loisach, Neckar, and Schwingbach catchments (see Table 1 methods). The colors of the lines and    labels on the graph indicate the nine dominant land use classes.

**3.3.2    Land use variation**

Similar to water physico-chemical variables, the variability in GHG concentrations and fluxes was

more strongly linked to catchment land use than seasonality (Table 2). In the Loisach catchment, $CO_2$
concentrations and fluxes were an order of magnitude higher for the ditch and stream sites that were dominated



by grassland land uses than forested-dominated sites (Fig. 3; Fig. 4; Table B3). $N_2O$ concentrations and fluxes in streams were also an order of magnitude higher in the grassland streams compared to the wetland and forested ones, with the latter functioning as occasional sinks for atmospheric $N_2O$ (Fig. 3; Fig. 4; Table B3). Wetland streams had higher $CH_4$ fluxes than the other stream sites (Fig. 3; Fig. 4; Table B3). Overall, ditches showed up to 14 times higher $CO_2$ and up to 850 folds higher $CH_4$ concentrations than the streams of Loisach catchment (Fig. A5; Table B3). In contrast, $N_2O$ concentrations in the ditches were highly variable, with both higher and lower than atmospheric concentrations over the sampling year (Fig. A2). $CH_4$ fluxes were two orders of magnitude higher in ditches than in streams (Fig. 3; Fig. 4; Table B3). Interestingly, the ditches were even more often $N_2O$ sinks than forests, which resulted in overall lowest $N_2O$ fluxes e.g. 10 times lower than the ones of grassland-dominated streams (Fig. 3; Table B3)

In the Neckar sub-catchments, $CO_2$, $CH_4$, and $N_2O$ concentrations and fluxes were 1-10 times higher in the streams located in cropland and settlement areas as compared to streams in forested areas (Fig. 3; Fig. 4; Fig. A5; Table B3). Generally, GHG concentrations and fluxes of streams in cropland and settlement areas further increased if sampling points were affected by wastewater inflows (Fig. 3; Fig. 4; Fig. A5; Table B3). For the latter, it is noteworthy that pronounced differences in wastewater characteristics existed in our study, even though the treatment procedures and the number of served households (80000) were comparable for the two wastewater treatment plants. Overall, the wastewater outflow in the Ammer catchment had higher TDN, DOC, $CH_4$ and $N_2O$ concentrations than the one in the Steinlach catchment (Table B1).

In contrast to the other two catchments, forested streams in the Schwingbach catchment had $CO_2$ and $CH_4$ concentrations and fluxes comparable to cropland and settlement-influenced streams within the catchment (Fig. 3; Fig. 4; Fig. A5; Table B3). However, $N_2O$ concentrations and fluxes were higher in streams with cropland and settlement influences than in forested streams (Fig. 3; Fig. 4; Fig. A5; Table B3).



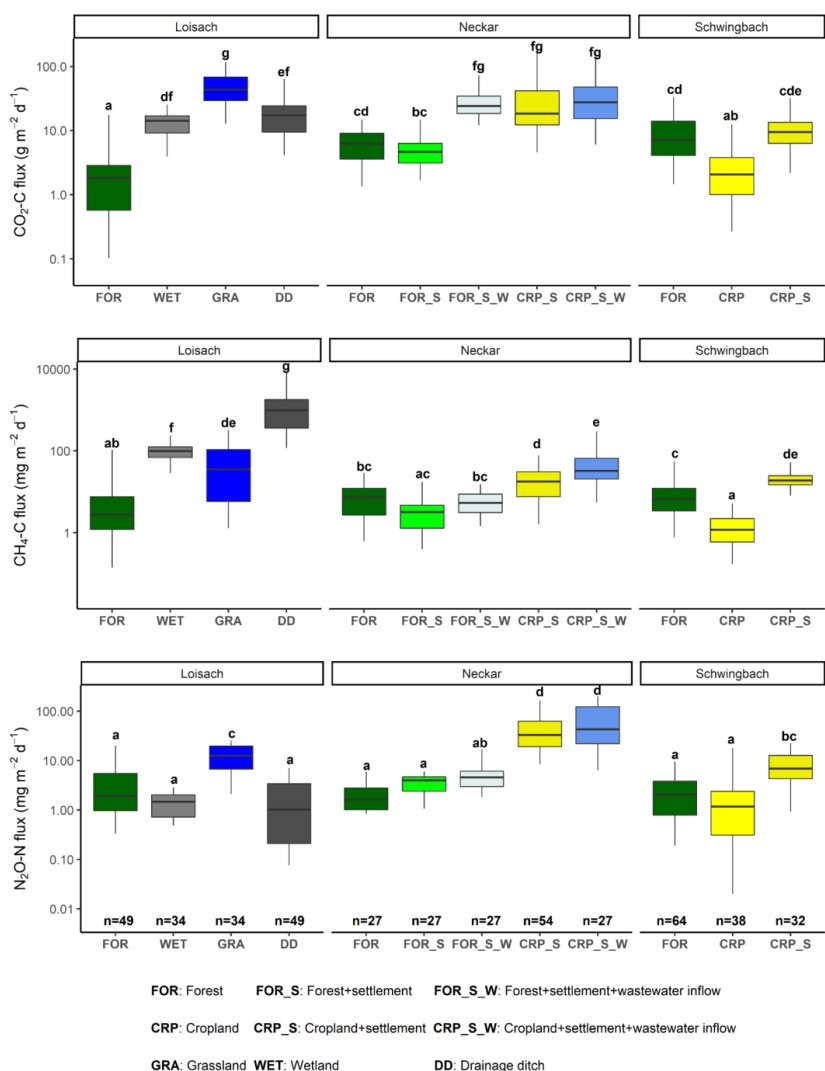

Fig. 4: Boxplots of $CO_2$, $CH_4$, and $N_2O$ fluxes in stream and ditch waters in the three catchments grouped by land
uses (see Table 1 methods). Letters on top of the boxplots represent significant differences ($p < 0.05$) amongst the
land use classes across the three catchments based on Tukey post-hoc analyses from the linear mixed-effects
models' results (Table 2).





### 3.4 Direct and indirect drivers of greenhouse gas concentrations

We used path analyses from SEMs based on all our dataset in order to explain how indirect factors such as upstream agricultural area, wastewater inflow and stream velocity controlled the spatial-temporal dynamics of GHG concentrations that drove the fluxes. The slopes parameter estimates from the SEMs revealed significant ($p<0.05$) interactions between the aforementioned indirect drivers and DO (% saturation), DOC mg $L^{-1}$ and $NO_3$-N mg $L^{-1}$, i.e. drivers that directly control *in situ* GHG concentrations (Fig. 5, Table B4). An increase in upstream agricultural area resulted in a ~45% increase in *in situ* $NO_3$-N concentrations. Wastewater inputs resulted in a ~22% increase in *in situ* $NO_3$ concentrations, while DOC concentrations were not significantly affected. DO decreased with increasing DOC concentrations, while $NO_3$-N concentrations followed an opposite pattern and increased with increasing DO concentrations (Fig 5).

$CO_2$ and $CH_4$ concentrations had a negative relationship with DO (Fig 5A-B), but $N_2O$ concentrations were not significantly related to DO (Fig 5C). Besides DO, $CO_2$ concentrations decreased by 17% with stream velocity, and increased by 18% with wastewater inflows and by 24% with increasing upstream agricultural area (Fig 5A). $CH_4$ concentrations also decreased by 16% with increasing stream velocity. However, the effect of wastewater inflows (+5%) or increased share of agricultural areas (+12%) on $CH_4$ concentrations was lower than for $CO_2$. Additionally, $CH_4$ concentrations also decreased by 31% with increasing $NO_3$-N concentrations (Fig. 5B). In contrast to $CO_2$ and $CH_4$, $N_2O$ concentrations increased by 40% with increasing $NO_3$-N concentrations, while the effect of stream velocity was of minor importance (-8%). Compared to $CH_4$ and $CO_2$, $N_2O$ concentrations in stream and river waters showed similar or stronger relationships to wastewater inflows (+18%) and upstream agricultural area (+34%) (Fig 5C). Overall, the best-fit SEMs explained 60, 66, and 46 % of the observed variances in $CO_2$, $CH_4$, and $N_2O$ concentrations, respectively (Table B4)



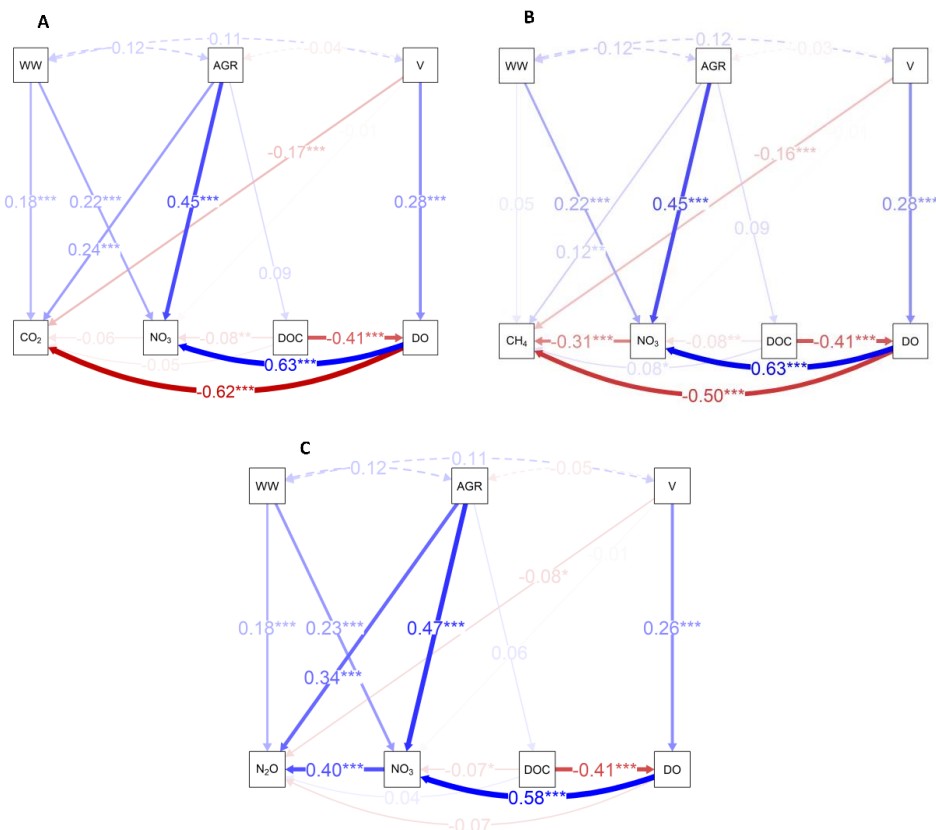

Fig. 5: Regression pathways predicting A) $Log_e$ $CO_2$ concentration µg-C $L^{-1}$, B) $Log_e$ $CH_4$ concentration µg-C $L^{-1}$ and C) $Log_e$ $N_2O$ concentration ng-N $L^{-1}$ across all sampling points and seasons from best-fit SEMs consisting of endogenous (DO, DOC, and $NO_3$-N) and exogenous variables (stream velocity (V), percentage agricultural area (AGR; grassland+cropland areas), and wastewater inflows (WW). The numbers on the lines represent standardized slope parameters, with significant relationships indicated by *. Solid lines represent actual fitted relationships, while dashed lines represent co-variances in the exogenous variables. Blue lines represent positive relationships and red represents negative relationships, with width and color intensity represting the strength of the relationships.

### 3.5    Annual areal fluxes

Based on global warming potential calculations, $CO_2$ dominated the annual GHG emissions across all headwater streams, with contributions ranging from 56 %–100%. The non-$CO_2$ gasses' contributions were much lower and ranged from 0–43% for $CH_4$ and 0–21% for $N_2O$ (Fig. 6). The highest contribution of $CH_4$ (43%) was found at ditch sampling points in the Loisach, while the highest $N_2O$ contributions (up to 21%) were observed at the cropland-influenced streams fed by wastewater inflows in the Neckar sub-catchments (Fig. 6). Overall, the



annual $CO_2$-equivalent emissions from anthropogenic-influenced streams were up to 20 times higher than from
natural forested and wetland streams (~71 kg $CO_2$ m$^{-2}$ yr$^{-1}$ vs. ~3 kg $CO_2$ m$^{-2}$ yr$^{-1}$ respectively; Fig. 6). Its also
noteworthy that the total annual GHG emission from oligotrophic forested streams in the Loisach catchment
were significantly lower than other forested catchments in the more human influenced Schwingbach and Neckar
sub-catchments (Fig. 6).
Regarding different discharge periods, high and medium discharge periods contributed up to 91 % to
total GHG emissions in anthropogenic-influenced streams, but only 4% in forested streams (Fig. 6). Overall, the
high and medium discharge periods contributed the most to the annual fluxes quantified in lower-order streams
(Strahler 1-2) and ditch sampling points, which were prevalent in the Loisach and Schwingbach sub-catchments
(Fig. 6B, C). The opposite was true for larger forested and cropland streams in the Neckar sub-catchment, where
higher annual flux contributions occurred primarily in the low discharge period (Fig. 6A). However, this pattern
did not hold true for cropland streams with the wastewater inflows in the same catchment, with the sites showing
an 82% increase in annual emissions during the high and medium discharge periods (Fig. 6 B, C).





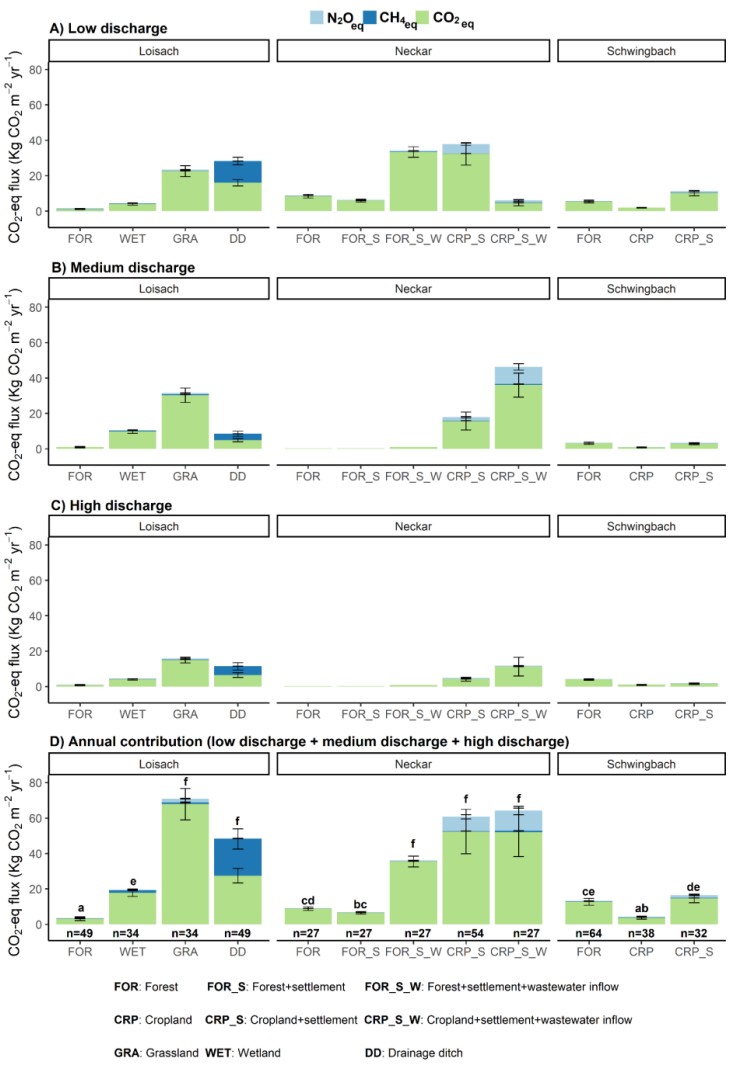


Fig. 6: Areal $CO_2$-equivalent fluxes (mean ±SE) grouped by GHG type for each land use class during A) low, B)
medium, and C) high discharge periods. D) represents the total annual fluxes by summing up contributions from
the three discharge periods. Letters on the bar graphs represent significant differences ($p<0.05$) in the annual
areal emissions amongst the land use classes across the three catchments based on Tukey post-hoc analyses from
the linear mixed-effects models' results (Table 2)



## 4    Discussion

### 4.1    Seasonal variability in GHG concentrations and fluxes

The GHG fluxes quantified from headwater streams and ditches in the three catchments in central and southern Germany add to the growing evidence that both aquatic ecosystems are significant net emitters of GHGs to the atmosphere. Seasonal trends in *in situ* GHG concentrations and fluxes were mainly linked to substrate availability (C and N), discharge and temperature, similar to previous studies on other streams in temperate climates (Dismore et al., 2013; Herreid et al., 2021). However, the observed seasonality of GHG fluxes from streams and ditches in our study was further impacted by land use across the three investigated catchments, with sub-catchments dominated by wetlands or forested land uses exhibiting lower seasonal variabilities than sub-catchments dominated by agricultural land use or affected by wastewater inflow (Fig. 3).

The low *in situ* $CO_2$ concentrations (< 100% saturation) during summer (Table B2) suggested elevated photosynthetic uptake within the streams and ditches, which is in line with the results of a recent meta-analysis on lotic ecosystems (Gómez-Gener et al., 2021). The decline in $CO_2$ concentrations in summer was most obvious at the non-forested stream sampling points, with higher canopy cover in the forested areas likely limiting *in situ* stream photosynthesis due to shading effects. We also found that stream ditch waters were oversaturated with $CO_2$ in autumn and winter. These seasons are characterized on the one hand by low discharge and low stream velocity, conditions which likely reduce degassing rates, and on the other hand by elevated *in situ* C metabolism, as supported by low DO concentration in autumn, which indicates respiratory $O_2$ consumption (e.g., Borges et al., 2018). We attribute the lack of seasonality in $CO_2$ fluxes (Table B2) to the compensatory effects of seasonally varying stream velocities and $CO_2$ source strengths. For example, high $CO_2$ concentrations and low gas transfer velocities in autumn and vice versa conditions in spring, resulted in comparable $CO_2$ fluxes amongst the two season (Table B2).

$N_2O$ concentrations also varied significantly across seasons, but the pattern differed from that of $CO_2$. In autumn, forested lower-order streams in the Loisach and Schwingbach catchments mainly showed $N_2O$ concentrations below atmospheric background concentrations and were temporary sinks of $N_2O$ (Fig. 3). This finding could be related to increased inputs of organic matter in these headwater catchments due to leaf fall, providing additional organic carbon for microbial metabolism in this period, which likely increased the demand for terminal electron acceptors such as $O_2$, $NO_3$, as well as $N_2O$. This conclusion is also supported by lowest DO and $NO_3$-N concentrations during autumn, which could suggest the dominance of complete denitrification in the streams (Quick et al., 2019). With decreasing temperatures towards winter, lower productivity and N demand within the streams resulted in the accumulation of $NO_3$-N, which seemed to favor internal $N_2O$ production as seen by the positive relationship between the two variables (Fig. 5C). The high sensitivity of the $N_2O$ reductase to low temperatures might have further supported elevated $N_2O$ concentration and fluxes during winter (e.g., Holtan-Hartwig et al., 2002). A similar finding of high winter $N_2O$ concentrations and fluxes was also found in other temperate streams, alluding to similar controls of temperature and nutrient availability (Herreid et al., 2021; Galantini et al., 2021). Thus, based on our results, winter periods can significantly contribute to annual $N_2O$ emission budgets. Yet, to the best of our knowledge, temperate studies covering the winter period are still scarce.



In contrast to $CO_2$ and $N_2O$, neither $CH_4$ concentrations nor fluxes showed any seasonal trends. Such a

finding is similar to what was found in a global meta-analysis (Stanley et al., 2016), where multiple controls
related to substrate availability, geomorphology and hydrology were shown to result in high spatial-temporal
variance of $CH_4$, thus masking any seasonal emission patterns.

### 4.2    Effect of human impacts on GHG concentrations and fluxes

Anthropogenic-influenced streams and ditches draining predominantly agricultural and settlement areas
showed higher $CO_2$-equivalent GHG emissions than forested streams (Fig. 6). Such a finding is similar to other
studies in the temperate region (e.g., Borges et al., 2018; Galantini et al., 2021). The high GHG emissions of
streams  and ditches in agricultural and settlement areas is likely due to elevated hydrological inflow (e.g., via
groundwater and interflow) of nitogen and labile carbon (Lambert et al., 2017, Mwanake et al., 2019) or
terrestrialy originating dissolved GHGs linked to lower vegetation cover compared to forested catchments (e.g.,
Mwanake et al., 2022). This interpretation could be supported by the significant positive relationships that we
found between percentage agriculture and stream $CO_2$, $CH_4$ and $N_2O$, as well as nitrate concentration, and a
positive trend for DOC (Figure 5).
Low DOC:DON ratios have been previously linked to more labile and less aromatic forms of dissolved
organic matter (DOM) (Sebestyen et al., 2008, O'Donnell et al., 2010). We found significantly lower DOC:DON
ratios in streams  and ditches in agricultural and settlement areas than in forested streams, suggesting that the
more bioavailable DOM in the human-influenced ecosystems, favored elevated GHGs production through
heterotrophic processes (e.g., Bodmer et al., 2016). Such differences in DOC:DON ratios were also found
amongst forested streams, with a decreasing trend from Loisach, Neckar to Schwingbach catchments, which may
also explain the differences in their GHG emissions (Fig. 6). The differences in the DOM bioavailability of
forested streams in the three catchments may suggest differences in DOM flowpaths during terrestrial-
groundwater-stream interactions. We contend that the moderately sloping streams of the Neckar and
Schwingbach catchments, likely had lower DOC:DON ratios due to longer water residence time and higher
contributions of groundwater inflow (e.g., Sebestyen et al., 2008)  than those in the steeper forested catchments
of the Loisach (Table B3). Distinct difference in water stable isotope signatures, i.e. the shift of precipitation vs.
stream water seasonality across the three catchments (data not shown), further supported the difference in water
residence times and their relationships with stream slope (e.g., Zhou et al., 2021).
In addition to land use influences, wastewater inflows into streams in agricultural and settlement areas
further increased GHG concentrations and fluxes. The two sampled wastewater effluents, which drained into the
Steinlach and Ammer streams of the Neckar sub-catchments, showed higher GHG concentrations than the
stream water upstream of the inflows (Fig. A5, Table B1), which mainly lead to increased GHG concentration
and fluxes also downstream of the wastewater inflows. This finding is similar to what was found in other
temperate studies comparing stream GHG concentration upstream and downstream of wastewater inflows (e.g.,
Marescaux et al., 2018; Aho et al., 2022). However, due to higher background GHG fluxes in the cropland than
the forested sub-catchments (Fig. 4), differences in the total GHG emissions before and after wastewater inflow
were more pronounced in the forested sub-catchments (Fig. 6). In addition to the pronounced differences in the
quality of the wastewater effluent (Table B1), this finding also shows the importance of background GHG fluxes
as influenced by catchment land use in assessing how wastewater inflows affect riverine GHG emissions.



Apart from land use influences, GHG fluxes from streams have been previously shown to decrease with stream order, as dissolved GHG inputs from groundwater and terrestrial sources also decrease (e.g., Hotchkiss et al., 2015, Turner et al., 2015, Mwanake et al., 2022). While our study design was not meant to explicitly asses stream order influences due to limited replication across a wide range of stream orders, we did find an opposite trend with stream order. For example, higher-order streams (stream orders> 5) in the Neckar sub-catchments dominated by croplands and with wastewater influences had mostly higher GHG fluxes than lower-order streams (stream orders < 3) in the Loisach and Schwingbach catchments. We therefore show a potential breakdown of stream order-GHG relationships in highly human-impacted lotic ecosystems, with disproportionately higher GHG emissions than in more natural ecosystems. We also show that significant nutrient and labile carbon supplies to higher-order streams, which create ideal conditions for GHG production and emission, may outweigh the physical disadvantages (e.g. lower surface area to volume ratio) of higher-order streams relative to lower-order streams.

Drainage ditches, characterized by low flow velocities and high DOC:DIN ratios, functioned as strong sources of $CO_2$ and $CH_4$ fluxes compared to streams. We assume that the low DO, high DOC, and low $NO_3$-N concentrations, along with high water retention times, supported high *in situ* $CH_4$ production rates in sediments, resulting in the overall highest contribution of $CH_4$ fluxes to total annual GHG emission budgets than streams(Figure 6). This interpretation is further supported by a significant negative relationship between $CH_4$ and DO, as well as $NO_3$-N concentrations, and a positive relationship with DOC concentrations, associations which have also been previously linked to *in situ* methane production in fluvial ecosystems (e.g., Baulch et al., 2011b; Schade et al., 2016). High $CH_4$ fluxes from drainage ditches were also found in other studies from both forested and wetland areas (e.g., Schrier-Uijl et al., 2011; Peacock et al., 2021b). Contrastingly, ditches were only weak sources or even sinks for atmospheric $N_2O$. This finding suggests $N_2O$ reduction to $N_2$ via complete denitrification, an interpretation already made in previous studies on lotic ecosystems (e.g., Baulch et al., 2011; Mwanake et al., 2019).

**4.3    Comparison of GHG flux magnitudes with other regional studies**

This study reported among the highest fluvial $CO_2$ emissions compared to other studies, with significant mean fluxes of up to 51 g-C m$^{-2}$ d$^{-1}$ (Table 4). We attribute this finding to moderate-steep slopes such as those quantified in the mountainous streams of the Loisach catchment or diffuse and point terrestrial dissolved GHG inputs from the more human-influenced Schwingbach and Neckar catchments, translating to higher fluvial GHG fluxes (Fig. 6). However, our high $CO_2$ fluxes are comparable with those quantified from other temperate streams in Canada and Switzerland with similar moderate-steep slopes and considerable dissolved $CO_2$ inputs from terrestrial landscapes (e.g., Mcdowell & Johnson. 2018; Horgby et al., 2019). The $CH_4$ fluxes from streams in this study are comparable with those previously found in temperate sub-catchments with similar land uses and altitudes, but are lower than those reported from permafrost streams in China (Zhang et al., 2020). Our $N_2O$ fluxes from cropland, settlement, and wastewater-influenced streams, are higher than those previously reported in a mixed landuse catchment (Schade et al., 2016), but our forest $N_2O$ fluxes are in the same range as those of other temperate forested streams (Aho et al., 2022). That said, these comparisons may be hampered, particularly for fluvial $N_2O$ fluxes, by the limited number of studies currently available (Table 4).



The average ditch $CH_4$ fluxes in this study are higher than those reported for forest and wetland draining

ditches in boreal and temperate regions (Table 4: Schrier-Uijl et al., 2011, Peacock et al., 2021b) and the global

mean provided by Peacock et al., (2021), which includes estimates from large canals. In contrast, $N_2O$ fluxes

from ditches in this study are lower than those quantified from $NO_3$-N-rich agricultural ditches in tropical and

temperate regions (Table 4: Harrison & Matson, 2002; Reay et al., 2003).






Table 4: Compilation of GHG emissions from temperate streams and ditches with comparable land use, climate, and altitude ranges.

| Land use/ land cover | Climate | Country | Geographical coordinates | Altitude (m) | Number of study reaches | Number of observations | Duration of study | CO₂-C flux (g m⁻² d⁻¹) Range | Mean | CH₄-C flux (mg m⁻² d⁻¹) Range | Mean | N₂O-N flux (mg m⁻² d⁻¹) Range | Mean | Reference |
|---|---|---|---|---|---|---|---|---|---|---|---|---|---|---|
| Forest/Loisach streams | Temperate | Germany | Table 1 | 616 – 2963 | 3 | 51 | Annual, 2022 | -0.05 – 17.4 | 2.4 | -0.4 – 164 | 10.5 | -9.2 – 20.3 | 1.1 | This study |
| Forest/Schwingbach streams | Temperate | Germany | Table 1 | 176 – 480 | 5 | 27 | Annual, 2022 | 0.08 – 33.4 | 9.5 | -0.02 – 54.6 | 9.9 | -1.6 – 9.6 | 2.1 | This study |
| Forest/Neckar rivers | Temperate | Germany | Table 1 | 319 – 610 | 1 | 80 | Annual, 2022 | 0.6 – 14.7 | 6.6 | 0.6 – 28.9 | 9.1 | -6.9 – 5.9 | 0.3 | This study |
| Forest+settlement/Neckar rivers | Temperate | Germany | Table 1 | 319 – 610 | 1 | 27 | Annual, 2022 | 0.6 – 14.9 | 4.9 | 0.4 – 17.3 | 3.9 | -7.7 – 6.0 | 2.2 | This study |
| Forest+settlement+wastewater/Neckar rivers | Temperate | Germany | Table 1 | 319 – 610 | 1 | 27 | Annual, 2022 | 12 – 71.7 | 28.3 | 1.4 – 15.2 | 6.5 | -2.8 – 17.1 | 3.9 | This study |
| Wetland/Loisach streams | Temperate | Germany | Table 1 | 616 – 2963 | 2 | 34 | Annual, 2022 | 2.8 – 25.2 | 13.3 | 17.2 – 237.5 | 101.7 | -1.6 – 2.9 | 0.8 | This study |
| Grassland/Loisach streams | Temperate | Germany | Table 1 | 616 – 2963 | 2 | 34 | Annual, 2022 | 6.1 – 115.9 | 50.7 | 1.3 – 324.5 | 73.2 | -0.8 – 25.5 | 12.4 | This study |
| Cropland/Schwingbach streams | Temperate | Germany | Table 1 | 176 – 480 | 3 | 48 | Annual, 2022 | 0.3 – 9.0 | 2.1 | 0.07 – 5.6 | 0.9 | -0.8 – 18 | 1.9 | This study |
| Cropland+settlement/Schwingbach streams | Temperate | Germany | Table 1 | 176 – 480 | 2 | 32 | Annual, 2022 | 0.6 – 32.0 | 8.6 | 0.6 – 52.6 | 14.9 | -0.8 – 22.4 | 6.5 | This study |
| Cropland+settlement/Neckar rivers | Temperate | Germany | Table 1 | 319 – 610 | 2 | 54 | Annual, 2022 | 4.5 – 181.3 | 39.1 | 1.6 – 77.5 | 21 | 8.4 – 165.7 | 46.9 | This study |
| Cropland+settlement+wastewater/Neckar rivers | Temperate | Germany | Table 1 | 319 – 610 | 1 | 27 | Annual, 2022 | 1.1 – 129.9 | 38.8 | 0.8 – 301.9 | 58.2 | 6.3 – 198.2 | 67.6 | This study |
| Forest streams | Temperate | USA | 43.0760° N, 107.2903° W | 1211 – 3311 | 1 | 253 | June – October, 2014 | 1.5 – 6.79 | 1.3 | 14.4 – 576 | 28.8 | | | Kuhn et al., 2017 |
| Forest streams | Temperate | USA | 40.2140° N, 105.4332° W | 2780 – 3505 | 2 | 11 | June – July, 2013 | 0.2 – 1.6 | 0.49 | 0.3 – 7.8 | 2.1 | | | Crawford et al., 2015 |
| Forest streams | Temperate | USA | 41.6032° N, 73.0877° W | 270 – 810 | 7 | 608 | 4 years, 2016 – 2019 | | | | | -0.4 – 29 | | Abo et al., 2022 |
| Forest streams | Temperate | USA | 41.6032° N, 73.0877° W | 270 – 810 | 7 | 608 | 4 years, 2016 – 2019 | -1.2 – 152 | 3.4 | 0.3 – 2870 | 28.7 | | | Abo et al., 2021 |
| Forest streams | Temperate | Canada | 49.270°N, 122.560°W | 1200 – 3050 | 1 | | Annual, 2017 | 8.7 – 1980 | 55.9 | | | | | Mcdowell and Johnson, 2018 |
| Mixed streams | Temperate | USA | 43.123°N, 71.1219°W | 165 – 348 | 3 | 37 | Annual, 2012 | | 0.4 – 1.1 | | 6 – 43.8 | | | Schade et al., 2016 |
| Mixed streams | Temperate | Switzerland | 46.1512° N, 7.0634°E | 1190 – 3051 | 1 | 300 | Annual, 2016 | 13.3 – 494.5 | 31 | | | -0.6 – 6.0 | | Horgby et al., 2019 |
| Mixed streams | Temperate | Europe | | | 34 | 107 | Annual, 2017 | -0.8 – 5.8 | | | | | | Attermeyer et al., 2021 |
| Grassland drainage ditches | Temperate | | Table 1 | 616 – 2963 | 3 | 50 | Annual,2022 | 2 – 63.3 | 13.7 | 116.6 – 7933 | 1532 | -0.8 – 7.1 | 1.2 | This study |
| Wetland drainage ditches | Temperate | Netherlands | 52.2200°N, 4.5300°E | 1 – 10 | 7 | 14 | June - July, 2009 | | 0.8 | | 606.6 | | | Schrier-Uijl et al., 2011 |
| Agricultural drainage ditches | Temperate | Scotland | 65.5000° N,3.2400° W | 58 – 68 | 10 | 22 | June - November, 2001 | | | | | 1.5 – 15.3 | 2.5 | Reay et al 2003 |



**Conclusions**

618   Compared to forests and wetlands, streams and ditches in agricultural and settlement areas were
characterized by significantly higher GHG fluxes with greater intra-annual variabilities. A combination of
wastewater inflows and agricultural land use resulted in the highest riverine $CO_2$ and $N_2O$ fluxes, particularly
during high discharge periods with substantial contributions of external dissolved GHGs. In general,
anthropogenic activities resulted in a potential breakdown of the expected decrease of the GHG source strengths
with increasing stream order, as higher-order streams in the Neckar sub-catchments with cropland and settlement
influences had higher concentrations and areal fluxes than small streams in the Loisach and Schwingbach
catchments. As most studies use stream order to upscale local and regional riverine fluxes, we show from our
results that caution must be taken in applying the methodology, particularly across catchments differing in land
use intensity.

628   In general, our findings indicate that future work should focus more on human-influenced headwater
stream ecosystems, as they contribute disproportionately large annual fluxes and are more temporally variable
than natural ones. Our study also found higher winter $N_2O$ fluxes, emphasizing the need for continuous sampling
regimes covering full years in order to reduce uncertainty in annual GHG emission estimates. Combining
continuous sampling regimes of all three biogenic GHGs ($CO_2$, $N_2O$, and CH4) across catchments with
contrasting land uses will further constrict riverine emissions and aid in developing targeted emission reduction
mitigation strategies.



**Appendices**
**Appendix A: Figures**

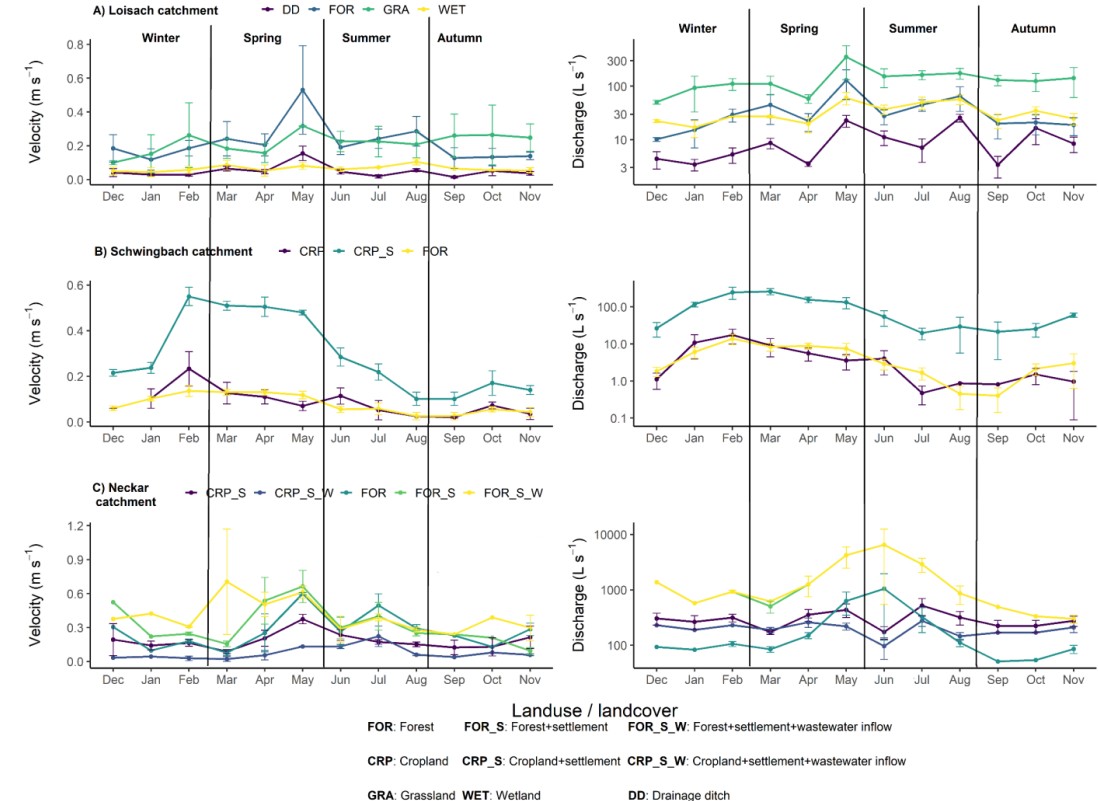


Fig. A1: Monthly mean ± SE velocity and discharge grouped by landuse / landcover classes in the A) Loisach,
B) Schwingbach and C) Neckar catchments.





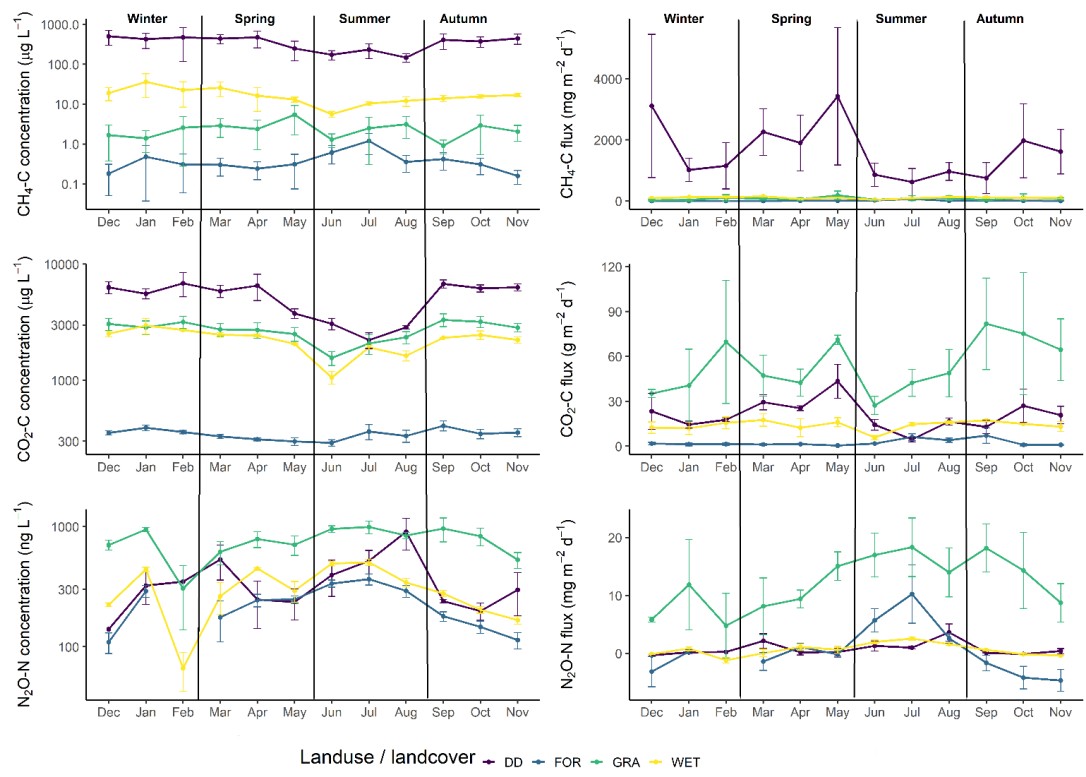

Fig. A2: Monthly mean ± SE CO$_2$, CH$_4$ and N$_2$O concentrations and fluxes at forested (FOR), wetland (WET), grassland (GRA) and ditch (DD) sites in the **Loisach** catchment (see Table 1 methods).



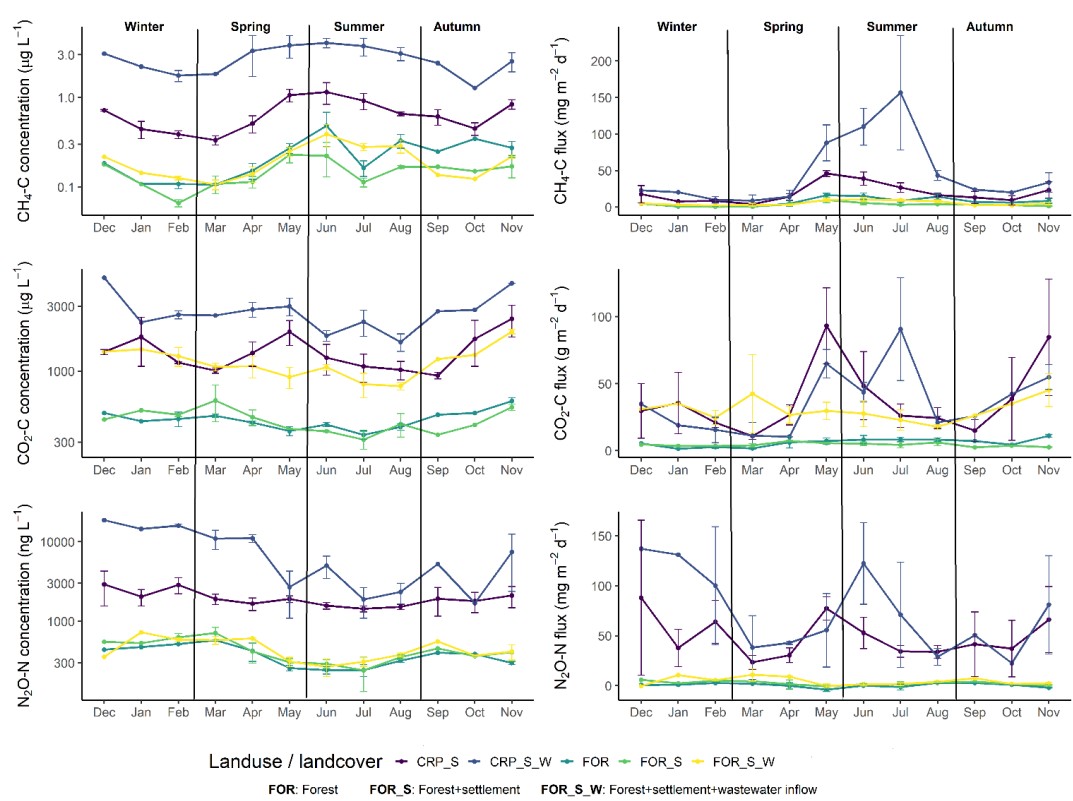

Fig. A3: Monthly mean ± SE $CO_2$, $CH_4$ and $N_2O$ concentrations and fluxes at forested (FOR), forested + urban (FOR_S), forested + urban + wastewater (FOR_S_W), cropland + urban (CRP_S) and cropland + urban + wastewater (CRP_S_W) sites in the **Neckar** catchment (see Table 1 methods).



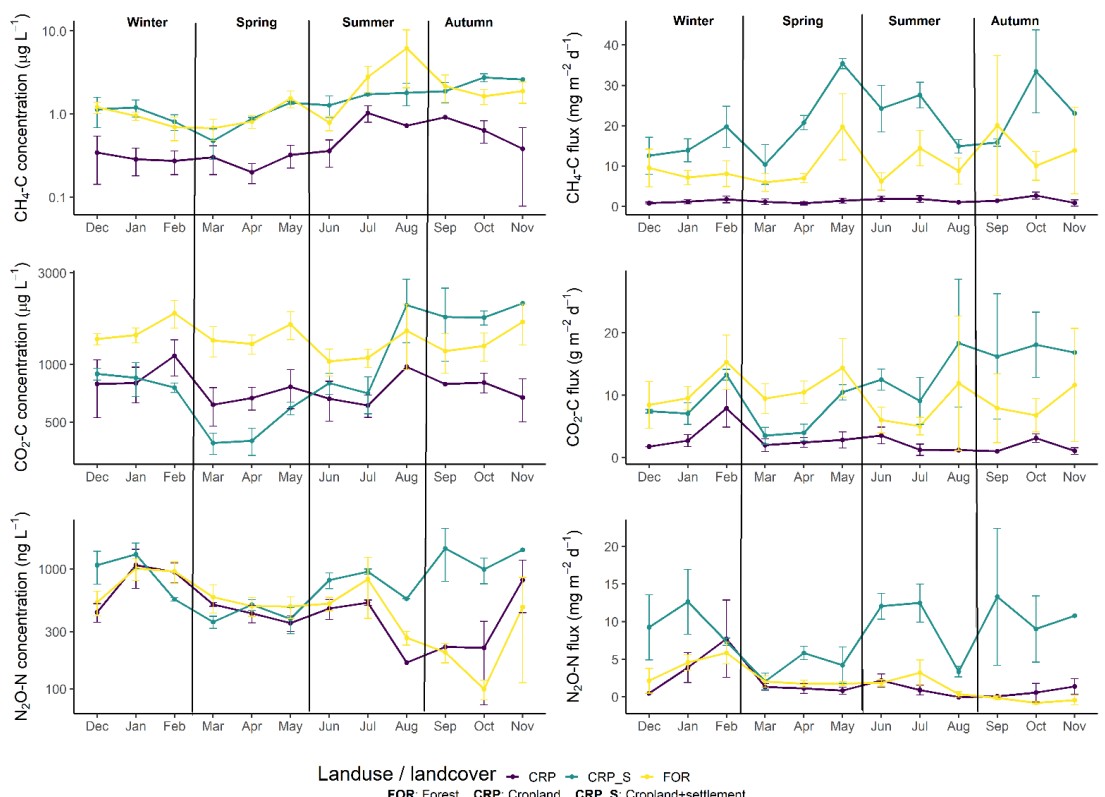



Fig. A4: Monthly mean ± SE $CO_2$, $CH_4$ and $N_2O$ concentrations and fluxes at forested (FOR), cropland (CRP)
and cropland + urban (CRP_S) sites in the **Schwingbach** catchment (see Table 1 methods).

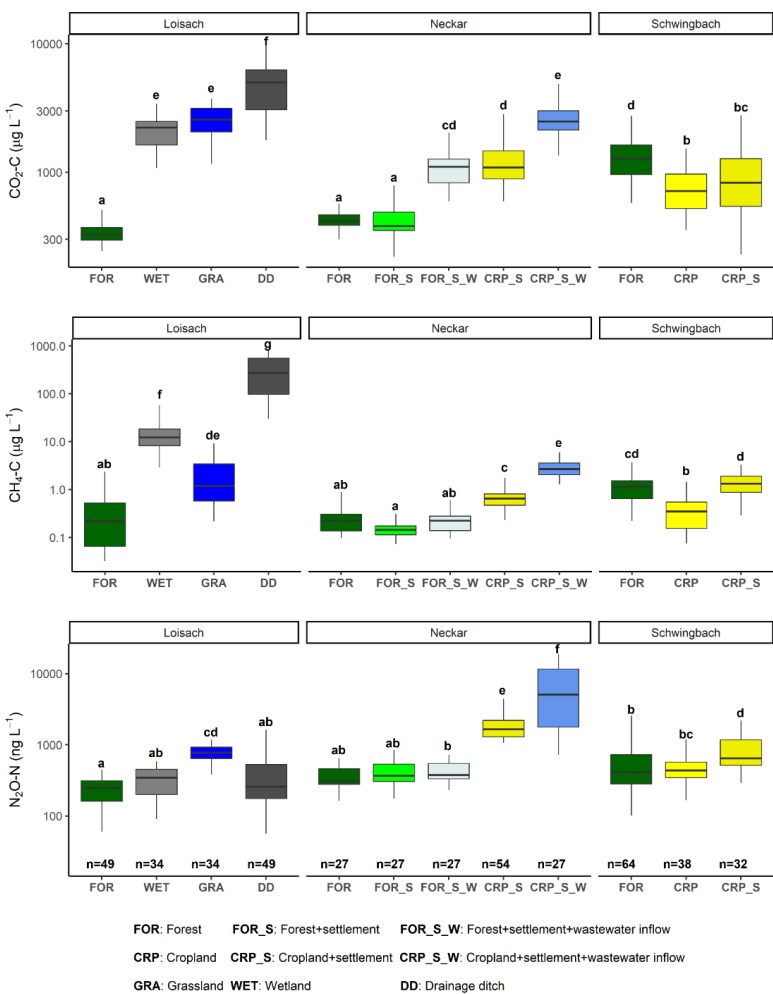

Fig. A5: Boxplots of $CO_2$, $CH_4$, and $N_2O$ concentrations in stream and ditch waters in the three catchments

grouped by dominating land uses (see Table 1 methods). Letters on top of the boxplots represent significant

differences ($p<0.05$) amongst the land use classes across the three catchments based on Tukey post-hoc analyses

from the linear mixed-effects models' results (Table 3).





**Appendix B: Tables**

Table B1: Annual means (+SE) of water chemistry variables and gas concentration measured in the effluents of
the Ammer (WWA) and Steinlach (WWS) wastewater treatment plants.

| Water quality variables and discharge | Wastewater effluent quality from inflow zones (Annual Mean ± SE) | |
| --- | --- | --- |
| | Ammer WWA | Steinlach WWS |
| Temperature (° C) | 13.85 ± 0.61 | 13.72 ± 0.65 |
| pH | 7.58 ± 0.07 | 7.37 ± 0.09 |
| DO (mg L$^{-1}$) | 6.01 ± 0.32 | 5.99 ± 0.34 |
| Specific Conductivity | 1017.96 ± 63.08 | 776.68 ± 63.48 |
| $NO_3$-N (mg L$^{-1}$) | 7.57 ± 0.6 | 6.33 ± 0.47 |
| $NH_4$-N (mg L$^{-1}$) | 0.14 ± 0.02 | 0.09 ± 0.03 |
| DOC (mg L$^{-1}$) | 6.8 ± 0.33 | 5.66 ± 0.58 |
| TDN (mg L$^{-1}$) | 8.43 ± 0.88 | 7.58 ± 0.88 |
| $CO_2$-C concentration (µg L$^{-1}$) | 4020.08 ± 192.75 | 4529.3 ± 224.37 |
| $CH_4$-C concentration (µg L$^{-1}$) | 2.13 ± 0.3 | 0.73 ± 0.09 |
| $N_2O$-N concentration (ng L$^{-1}$) | 9255.11 ± 1563.23 | 483.23 ± 61.35 |





Table B2: Seasonal means (+SE) of water physico-chemical variables, gas concentration and flux measured in
the Loisach, Neckar and Schwingbach catchments. Letters beside the means represent significant differences
(p<0.05) amongst the seasons across the three catchments based on Tukey post-hoc analyses from the linear
mixed-effects models' results (Table 2).

| | Summer | Autumn | Winter | Spring |
|---|---|---|---|---|
| Temperature (° C) | 14.04 ± 0.2 **d** | 9.83 ± 0.32 **c** | 5.55 ± 0.21 **a** | 8.38 ± 0.22 **b** |
| pH | 7.85 ± 0.03 **a** | 7.88 ± 0.04 **ab** | 7.98 ± 0.04 **b** | 7.96 ± 0.04 **ab** |
| DO (mg $L^{-1}$) | 8.71 ± 0.18 **a** | 8.55 ± 0.29 **a** | 9.63 ± 0.27 **b** | 9.85 ± 0.22 **b** |
| Specific Conductivity | 612.03 ± 21.8 **a** | 606.91 ± 28.44 **b** | 600.86 ± 32.62 **ab** | 555.63 ± 24.03 **a** |
| $NO_3$-N (mg $L^{-1}$) | 2.54 ± 0.22 **a** | 2.14 ± 0.29 **a** | 2.86 ± 0.28 **b** | 2.6 ± 0.22 **ab** |
| $NH_4$-N (mg $L^{-1}$) | 0.11 ± 0.01 **a** | 0.14 ± 0.02 **a** | 0.13 ± 0.02 **a** | 0.1 ± 0.01 **a** |
| TN (mg $L^{-1}$) | 2.9 ± 0.22 **a** | 2.49 ± 0.3 **a** | 3.01 ± 0.36 **b** | 3 ± 0.29 **ab** |
| DON (mg $L^{-1}$) | 0.5±0.07 **a** | 0.75±0.15 **a** | 1.56±0.26 **a** | 1.3±0.24 **a** |
| DOC (mg $L^{-1}$) | 4.37 ± 0.24 **a** | 4.26 ± 0.36 **a** | 4.1 ± 0.31 **a** | 4.66 ± 0.26 **a** |
| DOC:DIN | 11.45 ± 2.9 **b** | 7.21 ± 1.37 **ab** | 4.14 ± 0.75 **a** | 7.21 ± 1.81 **b** |
| DOC:DON | 103.91 ± 56.91 **a** | 183.33 ± 140.18 **a** | 13.19 ± 2.37 **a** | 28.33 ± 7.31 **a** |
| Stream velocity (m $s^{-1}$) | 0.18 ± 0.01 **ab** | 0.12 ± 0.01 **a** | 0.16 ± 0.01 **ab** | 0.24 ± 0.02 **b** |
| Discharge L $s^{-1}$ | 526.41 ± 171.4 **ab** | 86.25 ± 13.07 **a** | 157.3 ± 31.58 **ab** | 384.08 ± 96.29 **b** |
| $CO_2$ concentration (µg-C $L^{-1}$) | 1198.93 ± 71.66 **a** | 2222.22 ± 208.63 **c** | 1869.06 ± 185.95 **c** | 1666.03 ± 148.04 **b** |
| $CH_4$ concentration (µg-C $L^{-1}$) | 20.94 ± 5.36 **a** | 58.08 ± 17.8 **a** | 46.98 ± 18 **a** | 40.94 ± 13.03 **a** |
| $N_2O$ concentration (ng-N $L^{-1}$) | 816.06 ± 75.58 **ab** | 796.45 ± 169.08 **a** | 1691.19 ± 400.62 **b** | 1021.38 ± 185.45 **ab** |
| $k_{600}$ m$d^{-1}$ | 32.31 ± 3.09 **ab** | 22.71 ± 2.8 **a** | 24.54 ± 3.36 **ab** | 33.92 ± 3.42 **b** |
| $CO_2$ flux (mg-C $m^{-2} d^{-1}$) | 17008.98 ± 1876.63 **a** | 22710.21 ± 3422.95 **a** | 14836.51 ± 1835.54 **a** | 20592.21 ± 2563.97 **a** |
| $CH_4$ flux (mg-C $m^{-2} d^{-1}$) | 121.65 ± 30.93 **a** | 233.99 ± 84.4 **a** | 157.33 ± 73.04 **a** | 262.87 ± 89.31 **a** |
| $N_2O$ flux (mg-N $m^{-2} d^{-1}$) | 13.69 ± 2.22 **b** | 9.63 ± 2.86 **a** | 16.12 ± 4.05 **b** | 10.64 ± 2.11 **ab** |




Table B3: Annual mean ± standard errors of measured water physico-chemical variables, GHG concentration, and flux for land use classes in the Loisach (FOR: forest, WET: wetland, GRA: grassland, and DD: drainage ditches), the Neckar (FOR, FOR_S: forest+settlement, FOR_S_W: forest+settlement+wastewater inflow, CRP_S: cropland+settlement, and CRP_S_W: cropland+settlement+wastewater inflow, and the Schwingbach catchment (FOR, CRP: cropland and CRP_S). The number of observations in each land use class is represented by "n" in brackets. Letters beside the means represent significant differences (p<0.05) amongst the land use classes across the three catchments based on Tukey post-hoc analyses from the linear mixed-effects models' results (Table 2).

| | Loisach | | | | Neckar | | | | | Schwingbach | | |
| --- | --- | --- | --- | --- | --- | --- | --- | --- | --- | --- | --- | --- |
| | FOR (n=49) | WET (n=34) | GRA (n=34) | DD (n=49) | FOR (n=27) | FOR_S (n=27) | FOR_S_W (n=27) | CRP_S (n=54) | CRP_S_W (n=27) | FOR (n=64) | CRP (n=38) | CRP_S (n=32) |
| Temperature (° C) | 8 ± 0.5 a | 8.6 ± 0.4 ab | 9.5 ± 0.2 bd | 9 ± 0.5 bc | 10.44 ± 1.01 bd | 11.6 ± 1.01 de | 12.14 ± 0.85 ef | 11.7 ± 0.41 e | 13.06 ± 0.63 f | 9.7 ± 0.5 be | 9.9 ± 0.7 cdef | 9.8 ± 0.8 be |
| pH | 8.3 ± 0.01 de | 7.7 ± 0.01 b | 7.6 ± 0.01 b | 7.3 ± 0.01 a | 8.45 ± 0.05 e | 8.44 ± 0.05 e | 8.07 ± 0.05 cd | 8.13 ± 0.05 cd | 7.72 ± 0.08 b | 7.7 ± 0.01 b | 8 ± 0.01 c | 8 ± 0.1 c |
| DO (mg L⁻¹) | 11 ± 0.1 de | 8.3 ± 0.2 c | 7.4 ± 0.2 b | 4.2 ± 0.3 a | 11.49 ± 0.39 de | 11.57 ± 0.33 de | 10.62 ± 0.31 d | 11.65 ± 0.17 e | 8.3 ± 0.29 bc | 8.8 ± 0.1 c | 8.9 ± 0.1 c | 9 ± 0.1 c |
| Specific Conductivity | 365.1 ± 8.1 a | 436.9 ± 9.4 ab | 447.7 ± 2.3 bc | 484.9 ± 16.2 bcd | 738.51 ± 51.37 g | 582.07 ± 13.96 de | 700.87 ± 31.16 fg | 1116.86 ± 31.11 i | 971.46 ± 41.76 h | 389.7 ± 18.8 ab | 597.2 ± 13 ef | 566.4 ± 20.2 ce |
| NO₃-N (mg L⁻¹) | 0.8 ± 0.01 cd | 0.5 ± 0.01 b | 0.8 ± 0.01 cd | 0.1 ± 0.01 a | 0.57 ± 0.04 bc | 2.39 ± 0.13 e | 3.73 ± 0.29 ef | 6.74 ± 0.17 h | 7.18 ± 0.38 gh | 1.5 ± 0.1 d | 4.9 ± 0.4 fg | 2.3 ± 0.2 e |
| NH₄-N (mg L⁻¹) | 0.01 ± 0.001 ab | 0.01 ± 0.001 a | 0.01 ± 0.001 a | 0.3 ± 0.001 d | 0.07 ± 0.02 bc | 0.1 ± 0.01 cd | 0.11 ± 0.02 cd | 0.12 ± 0.01 d | 0.14 ± 0.02 d | 0.1 ± 0.01 d | 0.1 ± 0.01 d | 0.1 ± 0.01 d |
| TN (mg L⁻¹) | 0.7 ± 0.01 b | 0.4 ± 0.01 a | 0.7 ± 0.01 b | 0.9 ± 0.1 b | 0.73 ± 0.06 b | 2.3 ± 0.11 cd | 3.92 ± 0.3 ef | 6.57 ± 0.19 gh | 7.24 ± 0.53 h | 2.2 ± 0.2 c | 6.1 ± 0.5 fg | 3 ± 0.3 de |
| DON (mg L⁻¹) | 0.08±0.02 ab | 0.03±0.02 a | 0.06±0.03 acd | 0.45±0.04 cd | 0.3±0.05 d | 0.26±0.08 bd | 1.02±0.33 de | 2.76±0.48 e | 3.6±1.03 e | 0.65±0.11 d | 1.45±0.24 ce | 0.75±0.1 de |
| DOC (mg L⁻¹) | 2.9 ± 0.3 b | 1.8 ± 0.1 a | 1.5 ± 0.1 a | 9.5 ± 0.7 g | 5.9 ± 0.67 fg | 4.22 ± 0.35 bc | 4.12 ± 0.39 cdf | 3.47 ± 0.17 bd | 4.67 ± 0.23 ef | 4.8 ± 0.2 ef | 3.8 ± 0.1 cde | 4.7 ± 0.2 df |
| DOC:DIN | 4.23±0.46 ef | 4.48±0.73 ef | 2.06±0.22 d | 45.14±8.27 h | 13.19±2.32 g | 1.84±0.24 cd | 1.64±0.23 cd | 0.62±0.03 a | 0.85±0.09 ab | 5.89±1.1 f | 1.25±0.17 bc | 2.82±0.3 de |
| DOC:DON | 694.26±615.24 gl | 861.15±610.89 h | 93.39±57.03 cfh | 37.84±3.02 fg | 60.73±30.87 efg | 46.02±16.38 dfg | 18.06±10.65 acd | 5.68±1.9 a | 6.65±2.54 ab | 37.19±15.88 df | 9.02±2.67 ac | 13.13±2.9 bcde |
| Stream velocity (m s⁻¹) | 0.22±0.03 cd | 0.07±0.01 b | 0.22±0.02 cd | 0.05±0.01 a | 0.3±0.04 de | 0.34±0.04 ee | 0.4±0.04 e | 0.19±0.02 d | 0.09±0.02 ab | 0.09±0.01 ab | 0.1±0.01 ab | 0.29±0.03 de |
| Discharge Ls⁻¹ | 37.7 ± 7.3 c | 34.5 ± 3.2 cd | 142.1 ± 20.6 ef | 11.1 ± 1.4 b | 290.56 ± 109.66 ef | 2053.15 ± 705.38 g | 2117.15 ± 730.03 g | 318.55 ± 32.65 f | 196.67 ± 14.43 f | 5.4 ± 0.7 a | 5.4 ± 1.3 a | 94 ± 15.5 de |
| CO₂-C concentration (µg L⁻¹) | 337.9 ± 9.1 a | 2075.3 ± 107.8 e | 2559.5 ± 123.8 e | 4913.5 ± 285.4 f | 423.85 ± 14.6 a | 426.67 ± 24.18 a | 1093.04 ± 71.11 cd | 1372.92 ± 104.52 d | 2586.47 ± 191.08 e | 1350 ± 65.3 d | 748.9 ± 45.1 b | 1018.1 ± 117.6 bc |
| CH₄-C concentration (µg L⁻¹) | 0.4 ± 0.1 ab | 16.2 ± 2.2 f | 2.4 ± 0.4 de | 338 ± 37 g | 0.25 ± 0.03 ab | 0.15 ± 0.01 a | 0.23 ± 0.02 ab | 0.72 ± 0.06 c | 3.01 ± 0.25 e | 1.5 ± 0.2 cd | 0.4 ± 0.1 b | 1.5 ± 0.1 d |
| N₂O-N concentration (ng L⁻¹) | 240.9 ± 16.3 a | 323 ± 25.1 ab | 771.1 ± 42.2 cd | 431.3 ± 64.9 ab | 355.91 ± 24.26 ab | 405.94 ± 32.61 ab | 421.75 ± 28.5 b | 1846.46 ± 106.37 e | 6600.11 ± 1121.92 f | 569 ± 59.6 b | 540 ± 64.5 bc | 864.5 ± 89.4 d |
| k₆₆₀ ml⁻¹ | 80.9 ± 10.6 f | 10.5 ± 0.7 bc | 31.5 ± 3.1 df | 6.5 ± 0.6 a | 52.58 ± 5.1 f | 37.66 ± 3.56 ef | 43.41 ± 3.2 ef | 36.26 ± 2.54 ef | 19.95 ± 2.62 cd | 11.7 ± 1.1 ae | 7.1 ± 0.9 ab | 22.9 ± 1.8 de |
| CO₂-C flux (g m⁻² d⁻¹) | 2.39 ± 0.4 a | 13.33 ± 0.9 df | 50.71 ± 5.3 g | 20.52 ± 1.9 ef | 6.66 ± 0.8 cd | 4.89 ± 0.55 bc | 28.26 ± 2.8 fg | 39.16 ± 6.3 fg | 38.81 ± 6.5 fg | 9.54 ± 0.9 cd | 2.8 ± 0.4 ab | 10.96 ± 1.3 cde |
| CH₄-C flux (mg m⁻² d⁻¹) | 10.5 ± 4.3 ab | 101.7 ± 8.3 f | 73.2 ± 15.7 de | 1532.9 ± 244.8 g | 9.09 ± 1.5 bc | 3.88 ± 0.7 ac | 6.54 ± 0.81 bc | 21.09 ± 2.37 d | 58.23 ± 13.33 e | 9.9 ± 1.3 c | 1.5 ± 0.2 a | 21.5 ± 2 de |
| N₂O-N flux (ng m⁻² d⁻¹) | 1.1 ± 0.9 a | 0.8 ± 0.2 a | 12.4 ± 1.4 c | 1.2 ± 0.4 a | 0.32 ± 0.63 a | 2.2 ± 0.64 a | 3.96 ± 0.85 ab | 46.92 ± 5.02 d | 67.59 ± 11.34 d | 2.1 ± 0.3 a | 1.9 ± 0.6 a | 8.8 ± 1.1 bc |





Table B4: Indices highlighting the performance of the best-fit SEMs, which indicate significant interaction pathways of both direct and indirect drivers of in-situ GHG concentrations in temperate streams, rivers, and drainage ditches. The goodness of fit index (GFI), comparative fit index (CF1), Tucker lewis index, standardized root mean square residual (SRMR), and root means squared error of approximation (RMSEA) are measures of model goodness of fit, while the parsimony fit index (PNFI) compares the best-fit model to the theoretical-model.

| Greenhouse gas (GHG) | Performance indices for the best-fit SEMs | | | | | | Model comparison | |
| | GFI | CFI | TLI | SRMR | RMSEA | $r^2$ | PNFI Theoretical SEM | Best-fit SEM |
| --- | --- | --- | --- | --- | --- | --- | --- | --- |
| $CO_2$ concentration ($\mu$g-C L$^{-1}$) | 1.00 | 1.00 | 1.00 | 0.02 | <0.01 | 0.60 | 0.13 | 0.22 |
| $CH_4$ concentration ($\mu$g-C L$^{-1}$) | 1.00 | 1.00 | 1.00 | 0.02 | <0.01 | 0.66 | 0.13 | 0.22 |
| $N_2O$ concentration (ng-N L$^{-1}$) | 0.99 | 1.00 | 0.98 | 0.03 | 0.04 | 0.46 | 0.13 | 0.22 |

**Best-fit SEM structure:-**

1. Log GHG = DO + DOC + Log $NO_3$ + agricultural area + wastewater inflow + stream velocity

2. Log $NO_3$ = DO + agricultural area + wastewater inflow + stream velocity

3. DOC = agricultural area

4. DO = DOC + stream velocity

**Goodness of fit assesment:-** **GFI, CFI and TLI**: 0.90 - 0.95; Good fit and >0.95 Excellent fit

 **SRMR and RMSEA**: 0.05 - 0.08; Good fit and <0.05 Excellent fit



### Data availability

Our institute is currently constructing an own data infrastructure to host all available data from its scientist and link it to the main KIT database. We are working currently with our institute's data department to provide a publicly accessible DOI link for all our research data as directed by our institute's policy. We hope the data will be soon available before the review process completes. Reviewers are however invited to ask for the data anytime during the review process and it will be provided by the corresponding author via email.

### Author contribution

RM, RK, GG, CG, and KB designed the field experiments. RK, KB, TH, and LB provided the infrastructural funding and RM and EW did the field and laboratory work. RM did the statistical analysis, consulting with RK and GG. RM prepared the first draft manuscript, consulting with RK. All co-authors contributed to the final version.

### Acknowledgments

This research was funded by the German academic exchange service (DAAD) as part of RM's doctoral studies. Infrastructure for the research was provided by the TERENO Bavarian Alps/ Pre-Alps Observatory, funded by the Helmholtz Association and the Federal Ministry of Education and Research (BMBF). The authors would like to thank the entire laboratory staff at Karlsruhe Institute of Technology, Campus Alpin, Justus Liebig University Giessen, and the University of Tübingen for providing logistical support and supporting the gas and nutrient analyses. We also acknowledge the contributions of Alisson Kolar, Paul Levin Degott, Franz Weyerer, and Raphael Boehm during the field campaigns.

### Declaration of competing interest

The authors declare that they have no conflict of interest.



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
