# Peer review of "Anthropogenic activities significantly increase annual"

_EGUsphere, 2023_

## Author Comment (AC1)

**#Reviewer 1**

This manuscript presents a year-long dataset of GHG concentrations and fluxes from 5 headwater catchments in Germany from streams, agricultural ditches, and WWTP outflows. It identifies controls on GHG dynamics using mixed-effects models and structural equation models. In addition, it upscales flux rates to calculate annual emissions in terms of global warming potential. The main finding, that anthropogenically impacted streams have higher and more variable GHG concentrations and fluxes, was well supported. Overall, the manuscript presents results that will be an important contribution to our understanding of GHG emissions from inland waters and I find the analysis and results novel and worthy of publication.

I have a few suggestions, although they mostly minor and easy to address.

Abstract:

-I don't think that the analysis backs up the statements about separating in situ production of GHGs and direct inputs of GHGs (e.g., ln 27-28, 30-31). These statements should be removed or rephrased.

**Response:** Thank you for your critical comment and suggestion. We removed the initial sentences and rephrased them to clarify our meaning.

"Our findings also suggested that nutrient, labile-carbon, and dissolved GHG inputs from the agricultural and settlement areas may have supported these hotspots and hot-moments of fluvial GHG emissions."

-I think the authors should more clearly state that anthropogenically impacted streams have not only higher, but also \*more variable\* GHG emissions than natural streams in the abstract. (i.e., give some sort of variability stats)

**Response:** Thank you for the suggestion. We have added a sentence to the abstract to represent this finding better.

"Streams in agricultural-dominated catchments or with wastewater inflows had up to 10 times higher $CO_2$, $CH_4$, and $N_2O$ emissions, which were also more temporally variable (CV > 55%) than forested streams."

-I would consider mentioning some of the other main findings in the abstract (if possible within word count limits): 1) the break down of the expected stream-order patterns in impacted sites and 2) the finding that CO2 is the dominant contributor in terms of global warming potential

**Response:** Thank you for the suggestion. We have added several sentences to reflect both findings in the abstract.

"Overall, the annual emission from anthropogenic-influenced streams in $CO_2$-equivalents was up to 20 times higher (~71 kg $CO_2$ $m^{-2}$ $yr^{-1}$) than from natural streams (~3 kg $CO_2$ $m^{-2}$ $yr^{-1}$), with $CO_2$ fluxes accounting for up to 81 % of the annual emissions, while $N_2O$ and $CH_4$ accounted for up to 18 and 7 %, respectively. The positive influence of anthropogenic activities on fluvial GHG emissions also resulted in a breakdown of the expected declining trends of fluvial GHG emissions with stream size. Therefore, future studies should focus on anthropogenically perturbed streams, as their GHG emissions

are much more variable in space and time and can potentially introduce the largest uncertainties to fluvial GHG estimates"

Methods:

-I don't see temperature/seasonality or NH4 in the SEM results, even though these parameters are listed as input variables. Were they found insignificant and dropped? Please clarify.

**Response:** Thank you for your question. Temperature and $NH_4$ were removed from the SEMs as they were insignificant. We have now clarified this in the results section.

"In contrast to all other variables, water temperature and $NH_4$-N mg $L^{-1}$ did not contribute significantly (p-value>0.05) to the variance explained by the best-fit SEMs and were removed from the final path analyses (Table B4)."

Results:

F2. Consider using colors to represent major land-use classifications and shades to differentiate the sub-classifications. For example, crop, crop + settlement, and crop + settlement + WW inflow could be given different shades of the same color. Also, yellow is somewhat difficult to see on all these plots.

**Response:** Thank you for the suggestion. We have edited all the colors in our plots to reflect this suggestion.

[Figure]

**FOR**: Forest  **FOR_S**: Forest+settlement  **FOR_S_W**: Forest+settlement+wastewater inflow

**CRP**: Cropland  **CRP_S**: Cropland+settlement  **CRP_S_W**: Cropland+settlement+wastewater inflow

**GRA**: Grassland  **WET**: Wetland  **DD**: Drainage ditch

I don't have the background to fully assess how the SEM analysis was applied but it seems to make sense and F5 is great.

**Response:** Thank you for the compliment. We also found it practical in explaining how multivariate drivers interact to drive the intra-annual trends in GHG concentrations.

I find the conclusion that typical stream order patterns break down in anthropogenically impacted streams interesting (L573-584). However, I don't see the data presented in the results section. Please include it here (and perhaps add to the abstract as well).

**Response:** Thank you for the observation. We have added this information in the results section and also in the abstract.

**Abstract**

"The positive influence of anthropogenic activities on fluvial GHG emissions also resulted in a breakdown of the expected declining trends of fluvial GHG emissions with stream size."

**Results**

" In addition to land use effects, we also examined spatial variability in the GHG concentrations and fluxes linked to stream order differences. We found tendencies of higher $CO_2$, $CH_4$, and $N_2O$ concentrations and fluxes with increasing stream orders in the Schwingbach and Neckar catchments dominated by croplands and settlement areas. In contrast to the Neckar and Schwingbach catchments, GHG concentrations and fluxes in the more natural Loisach catchment decreased with stream order (Fig. A4). Comparing across catchments, higher stream orders (5&6) in the human-influenced Neckar catchment had higher or comparable GHG concentrations and fluxes than lower stream orders (1–3) in the Schwingbach and Loisach catchments(Fig. A4).                "

Discussion:

Consider discussing the result that CO2 was the main contributor when emissions of all three gases are converted to CO2 equivalences (F6). I found this result to be interesting and perhaps it deserves more attention in the manuscript.

**Response:** Thank you for the observation. We have added a sentence at the beginning of the discussion to indicate this finding and further expanded on the fact that an increase in upstream human activities increased the contributions of the $CH_4$ and $N_2O$ relative to $CO_2$.

"In agreement with previous studies, $CO_2$ accounted for most (>81 %) of the annual fluvial fluxes in $CO_2$ equivalents (e.g., Marescaux et al., 2018; Mwanake et al., 2022; Li et al., 2021). However, the presence of upstream agricultural and settlement areas seemed to alter these trends by reducing the contribution of $CO_2$ and increasing $N_2O$ and $CH_4$ contributions. The effects of the above anthropogenic activities on aquatic GHG dynamics were twofold. Drainage ditches were landscape hotspots for $CH_4$ emissions, while increasing upstream agricultural and settlement areas resulted in fluvial $N_2O$ hotspots."

---

## Author Comment (AC2)

**#Reviewer 2**

In general, this is a very nice study design and thorough sampling and analysis. I do not see any issues in that regard. While most comments are minor, it does seem that the introduction and discussion sections have received less attention, and are currently quite superficial in some sections, and overlook or mix up some concepts/terminology. So while I like this study overall, my comments below highlight the need for the authors to dig a bit deeper in their framing and interpretation of the work relative to other work in the field. I support this paper being published if the authors can fix these issues.

General comments:

Issue with lateral inputs in introduction: Up to line 62, nowhere in the paper to this point is lateral transfer of GHGs mentioned. It is especially important for CO2 in headwater systems, but in the drainage ditches, the GHG production in the wetlands themselves must be a huge fraction of the emisisons budget, no? The introduction needs to more thoroughly reflect this aspect of mechanistic control.

**Response:** Thank you for the critical observation and suggestion for improvement. We have edited the introduction's first paragraph to better explain the mechanisms driving GHG seasonal and spatial dynamics in headwaters. We have also acknowledged the contribution of terrestrial soils to GHG dynamics within drainage ditches in our introduction (See response to a later comment and suggestion)

"Several biogeochemical processes are responsible for GHG production and consumption within headwater ecosystems. $CO_2$ production is mainly attributed to the respiration of organic matter (Battin et al., 2008). Production of $CH_4$ occurs through methanogenesis, with carbon dioxide and acetic acid as substrates under anaerobic conditions (Stanley et al., 2016). Methane consumption is also possible through methanotrophy in oxygen-rich stream waters, producing $CO_2$ (Shelley et al., 2014). $N_2O$ is mainly a byproduct in nitrification (under aerobic conditions) or an intermediate product in denitrification (under anaerobic conditions), but it can also be reduced to $N_2$ in organic-rich and nitrate-poor ecosystems (Quick et al., 2019). Apart from instream biogeochemical production, GHG concentrations in headwater streams may also originate from external sources such as groundwater and terrestrial soils (e.g., Borges et al., 2015; Hotchkiss et al., 2015). These external sources are generally dominant during periods of heavy precipitation when the hydrological connectivity between the streams and their surrounding terrestrial landscape and groundwater is activated. Yet, partitioning the sources of these GHGs between in-situ production and external sources remains a challenge, as their contributions are mainly compounded and also vary widely depending on discharge conditions and the surrounding land use (e.g., Aho & Raymond, 2019; Borges et al., 2019; Mwanake et al., 2022)."

The overview pgph in the intro line 75 and on is pretty thin on mechanistic insight. Please integrate lateral inputs of GHG more carefully, aside from one sentence, it is all about internal production. How do storms and seasonal changes in hydrology alter the balance between catchment CO2 loading and internal production?

**Response:** Thank you for the critical observation and suggestion for improvement. We have added sentences in the introduction to indicate times where seasonal variabilities in discharge control in situ vs external GHG sources.

"Seasonality in precipitation regulates discharge, whereby heavy precipitation events or snowmelt during spring can result in high discharge events. At the same time, dry summers and winter periods are often characterized by lower discharge (e.g., Aho et al., 2022). Previous studies have shown that low discharge periods with longer water residence times favor instream GHG production processes (e.g., Borges et al., 2018; Mwanake et al., 2022). In contrast, high discharge periods with shorter water residence times are unfavorable to *instream* C and N cycling, resulting in the dominance of externally sourced GHGs from upstream terrestrial sources depending on the surrounding land use. For example, studies have found that during high discharge periods, streams draining wetlands show peak $CO_2$ and $CH_4$ concentrations (e.g., Aho et al., 2019; Borges et al., 2019), and pronounced $N_2O$ concentrations are found in streams of cropland-dominated catchments (e.g., Mwanake et al., 2022)."

Discussion section 4.3: This section should be expanded. We need a more thorough numerical comparison with other studies.  There needs to be a conclusion, what is special or new about your study compared to those papers? Anything unique here? What can you say overall about the land use sites vs non?  I think that more work needs to be done in this section to take what is a very nice dataset and analysis, and actually make it impactful in terms of the insights that you are providing the community.

**Response:** Thank you for the critical observation and the suggestions for improving our discussion. We have expanded our discussion section to include a first paragraph that reflects on our key findings from this study, complementing what is currently in the conclusion section. We have also added a comparison of our flux estimates with current global synthesis datasets but limited the site-specific comparisons to the temperate region, which better represents the climatic as well as land management practices of our study area.

Discussion

[revised manuscript text omitted]

Line by Line comments:

L 29-31 - is it in situ, or is the ditch draining a landscape that has a lot of GHG production? This would be adjacent source, not in situ in the ditch.

**Response:** Thank you for your questions. Both sources can be significant as the ditches also have deep sediment layers dominated by particulate OM, which may favor internal production too. We have edited the line to reflect the existence of both sources.

"Besides draining $CH_4$ and $CO_2$-rich terrestrial soils, drainage ditches are characterized by short water residence times, high organic loads, and highly variable $O_2$ levels, which can simultaneously support vigorous $CH_4$ and $CO_2$ production and, subsequently, higher fluxes."

L33-34 - but natural systems make up a huge fraction of the total global stream area, so if your goal is to simply scale the contribution of streams, then your reasoning is not accurate. I'd refocus the implications here and say something more generalizable about why emissions research in these impacted rivers are important.

**Response:** Thank you for your critical comment and suggestion. We have rephrased this line to reflect on the importance of studying human-influenced streams.

"Therefore, future studies should focus on anthropogenically perturbed streams, as their GHG emissions are much more variable in space and time and can potentially introduce the largest uncertainties to fluvial GHG estimates."

L37 – In this figure, No indirect effects on GHG in this diagram in anthropogenic domain. Nutrients, hydrology, etc will also modify emissions patterns indirectly.

**Response:** Thank you for the observation. The indirect effects are included in the above-ground runoff and point wastewater sources indicated by the blue arrows.

L43 - photochemical DOM processing too.

**Response:** Thank you for the suggestion. It's true that photochemical processing also leads to $CO_2$ production. However, we were only interested in the primary biogenic process, respiration, as we aimed to relate it to the substrate and oxic levels within the lotic ecosystems. We have edited the sentence to reflect this.

"Biogenic $CO_2$ production is mainly attributed to respiration of organic matter (Battin et al., 2008)."

L48 – conclusion sentence needed.

**Response:** Thank you for the suggestion. We have edited the paragraph to include a conclusion.

"Yet, partitioning the sources of these GHGs between in-situ production and external sources remains a challenge, as their contributions are mainly compounded and also vary widely depending on discharge conditions and the surrounding land use (e.g., Aho & Raymond, 2019; Borges et al., 2019; Mwanake et al., 2022)."

L58 – compared

**Response:** Redone.

L69 – numerical context needed.

**Response:** Thank you for the observation. We have added the numerical context of the contributions.

"For example, in a study of urban-impacted rivers in the Seine basin in France, Marescaux et al. (2018) found elevated $CO_2$, $CH_4$, and $N_2O$ concentrations and fluxes downstream of wastewater inflows, which disproportionately contributed up to 52 % of the basin-wide annual GHG fluxes."

L70 – in rivers linked to land use, or direct land use emissions? Be specific here.

**Response:** Thank you for your critical comment. We have rephrased the sentence to reflect stream emissions and not from the terrestrial landscape.

"Similar findings were also found in urban-impacted rivers in China, where GHG emissions were up to 14 times higher than those in other land uses (Zhang et al., 2021)."

L64-74 - here, integrating the important discussion points from key articles would strengthen your introduction. I list a few but there are also more:

Park JH, Nayna OK, Begum MS, Chea E, Hartmann J, Keil RG, Kumar S, Lu X, Ran L, Richey JE, Sarma VV. Reviews and syntheses: Anthropogenic perturbations to carbon fluxes in Asian river systems–concepts, emerging trends, and research challenges. Biogeosciences. 2018 May 17;15(9):3049-69.

Begum MS, Bogard MJ, Butman DE, Chea E, Kumar S, Lu X, Nayna OK, Ran L, Richey JE, Tareq SM, Xuan DT. Localized pollution impacts on greenhouse gas dynamics in three anthropogenically modified Asian river systems. Journal of Geophysical Research: Biogeosciences. 2021 May;126(5):e2020JG006124.

**Response:** Thank you for sharing the references. We have adjusted the paragraph to add more context and updated the suggested references.

"In fluvial ecosystems within settlement areas, point-source inflows of wastewater effluents have also been reported to alter natural GHG trends along the river continuum (Park et al., 2018). The wastewater effluent is either substrate-rich, favoring insitu GHG production, or GHG-rich, resulting in high riverine GHG emissions downstream of the inflow point (e.g., Marescaux et al., 2018; Begum et al., 2021; Zhang et al., 2021; Wang et al., 2022). For example, in a study of urban-impacted rivers in the Seine basin in France, Marescaux et al. (2018) found elevated $CO_2$, $CH_4$, and $N_2O$ concentrations and fluxes downstream of wastewater inflows, which disproportionately contributed up to 52 % of the basin-wide annual GHG fluxes. Similar findings were also found in urban-impacted rivers in China, where their GHG emissions were up to 14 times higher than those in other land uses (Zhang et al., 2021). Yet, studies on GHG emissions from urban-impacted fluvial ecosystems are still scarce, and therefore their contributions to riverine annual GHG budgets are not well constrained. Moreover, little is known about the cumulative effects of diffuse and point pollution sources on the magnitude of riverine GHG fluxes and whether the diffuse pollution sources exert longer-lasting controls on their fluxes than the point sources."

L72 – constrained

**Response:** Redone.

L74 – through this part of the intro, this terminology or contrasting of urbanization and land use is confusing. Do you mean point- and non-point pollution impacts? Or by land use do you mean agricultural land use specifically? Please clarify.

**Response:** Thank you for the critical comment. We meant point and non-point sources in this context. We have rephrased the sentence to generally reflect this view.

"Moreover, little is known about the cumulative effects of diffuse and point pollution sources on the magnitude of riverine GHG fluxes and whether the diffuse pollution sources exert longer-lasting controls on their fluxes than the point sources."

L89 - this point could be reworded. It is essentially saying that low sampling frequency does not capture short term variability in GHG cycling. This is obvious, so could you take this line of thinking a step further?

**Response:** Thank you for the critical comment. We reworded the sentence and took it a bit deeper into the gaps to be addressed in our study.

"The dynamic interactions between seasonality and land use discussed above indicate that less frequent measurements of riverine GHG concentrations and fluxes may fail to capture periods of elevated fluvial emissions at spatially hotspot areas, resulting in an underestimation of the annual emissions."

L91 (and more general) - this statement is not exactly defensible. People have been measuring stream GHG patterns for decades. Note that the oldest reference cited here is 2018. A deeper exploration of the literature here would be needed to pinpoint more specific unknowns about riverine emission patterns. At the same time, much of the issues in this pgph is already presented in the last one discussing sub-annual patterns. Consider just removing or really going much deeper.

**Response:** Thank you for the critical comment. We reworded the paragraph to outline better our contribution to the already existing knowledge pull.

"The dynamic interactions between seasonality and land use discussed above indicate that less frequent measurements of riverine GHG concentrations and fluxes may fail to capture periods of elevated fluvial emissions at spatially hotspot areas, resulting in an underestimation of the annual emissions. Yet, only a handful of studies in temperate streams have assessed the seasonal dynamics of GHG fluxes at sampling points with contrasting land uses (e.g., Marescaux et al., 2018; Borges et al., 2018; Herreid et al., 2021; Galantini et al., 2021), resulting in limitations when analyzing the mechanisms that drive either hot periods or hotspots of fluvial GHG fluxes. As climate change causes more extreme discharge conditions and as agricultural intensification and settlement areas continue to increase (Winkler et al., 2021), more studies that cover a wide array of land uses, discharge, and temperature conditions are needed to allow developing better mechanistic understanding of their effects on fluvial GHG dynamics by unraveling synergistic or antagonistic relationships amongst them. These increased process understanding will form the basis of future mechanistic modeling approaches, which are essential to predict better how fluvial GHG emissions will respond to future climate and

land use changes (Battin et al., 2023)."

L209 – detail standards used for GC calibration.

**Response:** Thank you for the critical comment. We have added in the methods the calibration standards used for the GC.

"The standards used for the GC calibration were 450, 800, 1000, 1500, 2000, and 3000 ppm for $CO_2$, 1, 2, 3, 4, 5, and 6 ppm for $CH_4$ and 0.4, 0.8, 1, 1.5, 2, and 3 ppm for $N_2O$."

L278 - Terminology "end" and "exogenous is misleading throughout, because things like water temperature are within the system, so to me are not exogenous. Why not call them 'substrate' and 'environmental conditions' or something? This comment applies throughout the paper where this framework is used.

**Response:** Thank you for your suggestion. The terminologies are generic with the SEM framework. However, as suggested, we have adopted substrate and environmental conditions to replace the endo and exogenous, respectively.

"Path analysis from structural equation models (SEMs, "lavaan" package in R version 4.1.1) was used to determine how environmental factors linked to seasonality and land use directly or indirectly influenced *instream* GHG production and consumption processes as well as external GHG sources, i.e., dissolved GHG inputs to the streams originating from either wastewater inflows or terrestrial landscapes which were not produced *in situ*. In brief, these SEMs were constructed based on causal relationships between environmental variables (interpreted as ultimate drivers of GHG concentrations) and substrate variables, which are affected by the environmental variables and also act as immediate drivers that affect GHG concentrations. Substrate variables in the models, which are known to influence *in situ* biogeochemical GHG production and consumption processes directly, included dissolved oxygen DO (% saturation), DOC (mg $L^{-1}$), $NH_4$-N (mg $L^{-1}$), and $NO_3$-N (mg $L^{-1}$) concentrations (Battin et al., 2008; Stanley et al., 2016; Quick et al., 2019). The environmental variables in the models, which influence *in situ* GHG concentrations either directly by facilitating dissolved GHG inputs or indirectly by controlling the substrate variables, were water temperature (°C) (a proxy for different seasons), stream velocity V (m $s^{-1}$), % upstream agricultural area for each sampling point (AGR: grassland + cropland area) and wastewater inflows (WW:  Boolean numbers, i.e., 1 for the presence of wastewater inflow and 0 for absence)."

L305 – fold

**Response**: Redone

L337 - throughout the water chem and GHG results sections, when something is significant or not, please report the test statistic in the text.

**Response**: Thank you for your critical observation and suggestion. We have included the test statistic in all our results in the main text. See examples below

"Seasonality had an overall significant effect (p <0.05) on stream velocities across all sampling points, with higher stream velocities observed in spring (0.24 ± 0.02 m $s^{-1}$) than in autumn (0.12 ± 0.01 m $s^{-1}$) (Table 2; Table B2)."

"DO was higher in winter and spring than in summer and autumn (p<0.001). NO₃-N and TDN concentrations were highest in winter and lowest in autumn and summer (p<0.01), while NH₄-N, DOC, and DON showed no significant seasonal variation (p>0.05; Table 2; Table B2)."

L407 - either mention that the scales are not the same between the 3 columns, or preferably log transform the Y axis and use a consistent scale to facilitate comparisons between catchments.

**Response**: Thank you for your critical observation and suggestion. We have redone the figure to make the scale more consistent.

[Figure]

L460 - the light blue and red numbers and lines are not easy to see, reconfigure the plot to make those darker or something.

**Response**: Thank you for your critical observation and suggestion. We have redone the figure to colors to be better visible.

[Figure]

L477 – It is

**Response**: Thank you for your critical observation. We have rephrased the statement to make it clearer.

"Overall, the annual $CO_2$-equivalent emissions from anthropogenic-influenced streams (~71 kg $CO_2$ m$^{-2}$ yr$^{-1}$ ) were up to 20 times higher than from natural forested streams (~3 kg $CO_2$ m$^{-2}$ yr$^{-1}$; Fig. 6)."

L517 – seasons

**Response**: Redone

L598 - cite the relevant reviews and syntheses here

**Response**: Thank you for the suggestion. We have added three references with globally synthesized data in our comparison (Hu et al., 2016, Stanley et al., 2016; Li et al., 2021)

" This study's daily $CH_4$ and $N_2O$ diffusive flux ranges from both streams and ditches are mostly within the same order of magnitude as those previously reported in global synthesis studies (Table 3: Hu et al., 2016; Stanley et al., 2016). In contrast, this study reported among the highest fluvial $CO_2$ emissions compared to other regional and global studies, with significant mean fluxes of up to 51 g-C $m^{-2}$ $d^{-1}$ (Table 3)."

---

## Author Comment (AC3)

**#Reviewer 1**

This manuscript presents a year-long dataset of GHG concentrations and fluxes from 5 headwater catchments in Germany from streams, agricultural ditches, and WWTP outflows. It identifies controls on GHG dynamics using mixed-effects models and structural equation models. In addition, it upscales flux rates to calculate annual emissions in terms of global warming potential. The main finding, that anthropogenically impacted streams have higher and more variable GHG concentrations and fluxes, was well supported. Overall, the manuscript presents results that will be an important contribution to our understanding of GHG emissions from inland waters and I find the analysis and results novel and worthy of publication.

I have a few suggestions, although they mostly minor and easy to address.

Abstract:

-I don't think that the analysis backs up the statements about separating in situ production of GHGs and direct inputs of GHGs (e.g., ln 27-28, 30-31). These statements should be removed or rephrased.

**Response:** Thank you for your critical comment and suggestion. We removed the initial sentences and rephrased them to clarify our meaning.

"Our findings also suggested that nutrient, labile-carbon, and dissolved GHG inputs from the agricultural and settlement areas may have supported these hotspots and hot-moments of fluvial GHG emissions."

-I think the authors should more clearly state that anthropogenically impacted streams have not only higher, but also *more variable* GHG emissions than natural streams in the abstract. (i.e., give some sort of variability stats)

**Response:** Thank you for the suggestion. We have added a sentence to the abstract to represent this finding better.

"Streams in agricultural-dominated catchments or with wastewater inflows had up to 10 times higher $CO_2$, $CH_4$, and $N_2O$ emissions, which were also more temporally variable (CV > 55%) than forested streams."

-I would consider mentioning some of the other main findings in the abstract (if possible within word count limits): 1) the break down of the expected stream-order patterns in impacted sites and 2) the finding that CO2 is the dominant contributor in terms of global warming potential

**Response:** Thank you for the suggestion. We have added several sentences to reflect both findings in the abstract.

"Overall, the annual emission from anthropogenic-influenced streams in $CO_2$-equivalents was up to 20 times higher (~71 kg $CO_2$ m$^{-2}$ yr$^{-1}$) than from natural streams (~3 kg $CO_2$ m$^{-2}$ yr$^{-1}$), with $CO_2$ fluxes accounting for up to 81 % of the annual emissions, while $N_2O$ and $CH_4$ accounted for up to 18 and 7 %, respectively. The positive influence of anthropogenic activities on fluvial GHG emissions also resulted in a breakdown of the expected declining trends of fluvial GHG emissions with stream size. Therefore, future studies should focus on anthropogenically perturbed streams, as their GHG emissions

are much more variable in space and time and can potentially introduce the largest uncertainties to fluvial GHG estimates"

Methods:

-I don't see temperature/seasonality or NH4 in the SEM results, even though these parameters are listed as input variables. Were they found insignificant and dropped? Please clarify.

**Response:** Thank you for your question. Temperature and $NH_4$ were removed from the SEMs as they were insignificant. We have now clarified this in the results section.

"In contrast to all other variables, water temperature and $NH_4$-N mg $L^{-1}$ did not contribute significantly (p-value>0.05) to the variance explained by the best-fit SEMs and were removed from the final path analyses (Table B4)."

Results:

F2. Consider using colors to represent major land-use classifications and shades to differentiate the sub-classifications. For example, crop, crop + settlement, and crop + settlement + WW inflow could be given different shades of the same color. Also, yellow is somewhat difficult to see on all these plots.

**Response:** Thank you for the suggestion. We have edited all the colors in our plots to reflect this suggestion.

[Figure]

**FOR**: Forest    **FOR_S**: Forest+settlement    **FOR_S_W**: Forest+settlement+wastewater inflow

**CRP**: Cropland    **CRP_S**: Cropland+settlement    **CRP_S_W**: Cropland+settlement+wastewater inflow

**GRA**: Grassland    **WET**: Wetland    **DD**: Drainage ditch

I don't have the background to fully assess how the SEM analysis was applied but it seems to make sense and F5 is great.

**Response:** Thank you for the compliment. We also found it practical in explaining how multivariate drivers interact to drive the intra-annual trends in GHG concentrations.

I find the conclusion that typical stream order patterns break down in anthropogenically impacted streams interesting (L573-584). However, I don't see the data presented in the results section. Please include it here (and perhaps add to the abstract as well).

**Response:** Thank you for the observation. We have added this information in the results section and also in the abstract.

**Abstract**

"The positive influence of anthropogenic activities on fluvial GHG emissions also resulted in a breakdown of the expected declining trends of fluvial GHG emissions with stream size."

**Results**

" In addition to land use effects, we also examined spatial variability in the GHG concentrations and fluxes linked to stream order differences. We found tendencies of higher $CO_2$, $CH_4$, and $N_2O$ concentrations and fluxes with increasing stream orders in the Schwingbach and Neckar catchments dominated by croplands and settlement areas. In contrast to the Neckar and Schwingbach catchments, GHG concentrations and fluxes in the more natural Loisach catchment decreased with stream order (Fig. A4). Comparing across catchments, higher stream orders (5&6) in the human-influenced Neckar catchment had higher or comparable GHG concentrations and fluxes than lower stream orders (1–3) in the Schwingbach and Loisach catchments(Fig. A4).                "

Discussion:

Consider discussing the result that CO2 was the main contributor when emissions of all three gases are converted to CO2 equivalences (F6). I found this result to be interesting and perhaps it deserves more attention in the manuscript.

**Response:** Thank you for the observation. We have added a sentence at the beginning of the discussion to indicate this finding and further expanded on the fact that an increase in upstream human activities increased the contributions of the $CH_4$ and $N_2O$ relative to $CO_2$.

"In agreement with previous studies, $CO_2$ accounted for most (>81 %) of the annual fluvial fluxes in $CO_2$ equivalents (e.g., Marescaux et al., 2018; Mwanake et al., 2022; Li et al., 2021). However, the presence of upstream agricultural and settlement areas seemed to alter these trends by reducing the contribution of $CO_2$ and increasing $N_2O$ and $CH_4$ contributions. The effects of the above anthropogenic activities on aquatic GHG dynamics were twofold. Drainage ditches were landscape hotspots for $CH_4$ emissions, while increasing upstream agricultural and settlement areas resulted in fluvial $N_2O$ hotspots."

Conclusion:

L620 - It seems like CH4 is also higher in the anthropogenically impacted sites?

**Response:** Thank you for the observation. Yes. Methane fluxes were also higher in human-influenced streams. We have edited the conclusion to reflect this view.

"Streams and ditches in agricultural and settlement areas were characterized by significantly higher GHG fluxes with more significant intra-annual variabilities than forests and wetlands. A combination of wastewater inflows and agricultural land use resulted in the highest fluvial $CO_2$, $CH_4$, and $N_2O$ fluxes, particularly during high discharge periods with substantial external dissolved GHGs."

Minor:

L40 – "contributors to global greenhouse gas budgets" or "contributors of greenhouse gases"

**Response:** Thank you for the observation. Rivers are contributors to global greenhouse gas budgets. We have edited the statement to make it clearer.

"Streams and rivers cover only a small fraction of the earth's land surface (0.4%; Allen et al., 2018), yet they are significant contributors to global greenhouse ($CO_2$, $CH_4$, and $N_2O$) budgets, emitting approximately 7.6 (6.1–9.1) Pg-$CO_2$ equivalent into the atmosphere per year. (Li et al., 2021)."

L41 – Li et al., 2021 citation only refers to headwater streams

**Response:** Thank you for the question. The study also included the most recent cumulative GHG emissions from headwaters and large rivers, and that's why we used it to get an idea of the contributions of fluvial GHG fluxes to global budgets.

L52 – "in situ N2O production"

**Response:** Thank you for the observation. We have made the change in the text to make the sentence clearer.

"Elevated hydrological inputs of dissolved GHGs, nutrients, and labile carbon to streams from fertilized croplands have been shown to increase their $N_2O$ (e.g., Beaulieu et al., 2009), $CO_2$ (e.g., Bodmer et al., 2016; Borges et al., 2018), and $CH_4$ fluxes (e.g., Mwanake et al., 2022), by favoring *instream* GHG production processes and also ensuring steady supplies in periods of low in-situ biogeochemical production."

L70 – "where"

**Response:** Redone.

"Similar findings were also found in urban-impacted rivers in China, where their GHG emissions were up to 14 times higher than those in other land uses (Zhang et al., 2021)."

L260 – "May 31"

"Fixed effects in the models consisted of land use classes in each catchment (Table 1) and seasons: summer June 1–August 31, autumn September 1–November 30, winter December 1–February 28, and spring March 1–May 31."

L369 – consider using uatm for N2O, instead of natm, as it is more commonly used

**Response:** Thank you for the suggestion. We agree that uatm data units are commonly used. However, we used nano (uatm/1000) to represent better the low $N_2O$ concentrations measured in the forests and drainage ditches.

L375-376 - It looks like negative fluxes are actually presented in F3.

**Response:** Thank you for the observation. We indeed had negative fluxes. However, they were only limited throughout the year, hence the use of the word "mostly" to represent the majority of the fluxes, which were positive and dominated the annual emissions

---

## Author Response (AR2)

**Associate editor's comments**

Dear authors,
thank you for the revised version of your MS. I believe it is now of the quality required for publication, although I think reference to the paper of Deirmendjian et al. 2019 https://doi.org/10.1016/j.scitotenv.2019.01.152 could be helpful for the interpretation of your data at some places in the discussion, as it provides some additional insights on the transfer mechanisms of CO2 and CH4 from groundwaters to first order streams in crops and forest ecosystems. Please tell me if my suggestion is relevant or not, so we can complete the editorial process.
With best regards
Gwen Abril, BG associate editor

**Response:** Thank you for your overall comment and the reference suggestion. It was beneficial to explain some of our GHG seasonal trends in forested and cropland catchments, and we have now included them in the revised version.

"The decline in $CO_2$ concentrations in summer was most apparent at the non-forested stream sampling points, with higher canopy cover in the forested areas likely limiting *in situ* stream photosynthesis due to shading effects. These non-forested sites also had higher instream dissolved inorganic nitrogen concentrations, nutrient conditions previously shown to favor macrophyte photosynthetic uptake of $CO_2$, resulting in lower *in situ* stream $CO_2$ concentrations (Deirmendjian et al., 2019)."

"The high GHG emissions of streams and ditches in agricultural and settlement areas are likely due to elevated hydrological inflow (e.g., via groundwater and interflow) of nitrogen and labile carbon (e.g., Lambert et al., 2017; Deirmendjian et al., 2019; Mwanake et al., 2019) or terrestrially originating dissolved GHGs linked to lower vegetation cover compared to forested catchments (e.g., Deirmendjian et al., 2019; Mwanake et al., 2022). This interpretation could be supported by the significant positive relationships that we found between percentage agriculture and stream $CO_2$, $CH_4$, and $N_2O$, as well as nitrate concentration and a positive trend for DOC (Figure 5)."